# COVID-19 pandemic and Farr's law: A global comparison and prediction of outbreak acceleration and deceleration rates

**Kevin Pacheco-Barrios**[1,2], **Alejandra Cardenas-Rojas**[1☯], **Stefano Giannoni-Luza**[1☯], **Felipe Fregni**[1,3]*

**1** Spaulding Research Institute, Spaulding Rehabilitation Hospital and Massachusetts General Hospital, Harvard Medical School, Boston, Massachusetts, United States of America, **2** Unidad de Investigación para la Generación y Síntesis de Evidencias en Salud, Universidad San Ignacio de Loyola, Lima, Peru, **3** Department of Epidemiology, Harvard T.H. Chan School of Public Health, Boston, Massachusetts, United States of America

☯ These authors contributed equally to this work.
* ffregni@hsph.harvard.edu

**Data Availability Statement:** All relevant data are within the manuscript and its Supporting Information files.

## Abstract

The COVID-19 outbreak has forced most of the global population to lock-down and has put in check the health services all over the world. Current predictive models are complex, region-dependent, and might not be generalized to other countries. However, a 150-year old epidemics law promulgated by William Farr might be useful as a simple arithmetical model (percent increase [R1] and acceleration [R2] of new cases and deaths) to provide a first sight of the epidemic behavior and to detect regions with high predicted dynamics. Thus, this study tested Farr's Law assumptions by modeling COVID-19 data of new cases and deaths. COVID-19 data until April 10, 2020, was extracted from available countries, including income, urban index, and population characteristics. Farr's law first ($R_1$) and second ratio ($R_2$) were calculated. We constructed epidemic curves and predictive models for the available countries and performed ecological correlation analysis between $R_1$ and $R_2$ with demographic data. We extracted data from 210 countries, and it was possible to estimate the ratios of 170 of them. Around 42·94% of the countries were in an initial acceleration phase, while 23·5% already crossed the peak. We predicted a reduction close to zero with wide confidence intervals for 56 countries until June 10 (high-income countries from Asia and Oceania, with strict political actions). There was a significant association between high $R_1$ of deaths and high urban index. Farr's law seems to be a useful model to give an overview of COVID-19 pandemic dynamics. The countries with high dynamics are from Africa and Latin America. Thus, this is a call to urgently prioritize actions in those countries to intensify surveillance, to re-allocate resources, and to build healthcare capacities based on multi-nation collaboration to limit onward transmission and to reduce the future impact on these regions in an eventual second wave.

**Funding:** FF is funded by NIH grant R01 AT009491-01A1 (https://www.nih.gov). The funders had no role in study design, data collection and analysis, decision to publish, or preparation of the manuscript.

## Introduction

Covid-19 has been a global public health crisis for different reasons. This pandemic has had a rapid global spread, and in five months, 210 countries were affected by hundreds or thousands of cases and deaths [1]. Also, there was a lack of proper preparation, and it truly stressed the health care system, especially in countries where the incidence was higher, for instance, in China, Iran, and Italy [1].

One of the challenges in this crisis was the lack of good prediction models. In fact, a recent review of 27 studies analyzing 31 different models concluded that models have overall poor predictability and indeed should not be used to drive clinical care decisions [2]. There were several complex models limited to specific geographical regions that give us a partial perspective of this pandemic. It could also lead us to overestimate the number of cases and deaths when we try to replicate the models in other regions [3–6]. One can say that this would be theoretically beneficial as it would overprepare a health system for a rapid surge in cases, and thus, if the numbers are proven to be smaller, that would not have a large detrimental effect [7].

Another challenge in the COVID-19 epidemics is to design models for a condition with limited data. However, a similar scenario may repeat with another unknown or less studied agent. There have been simple prediction models that could have been best used to understand the numbers and dynamics of COVID-19 pandemic easily but from a planetary standpoint due to its accessible computation. In this article, we will discuss a simple and elegant method to forecast epidemic dynamics proposed almost 200 years ago by William Farr.

Farr's law stated that the epidemic's dynamic could be described as the relation of two arithmetic ratios. The first ratio ($R_1$) represents the change of cases or deaths comparing one time against the immediately before time. The second ratio ($R_2$) measures the rate of change of $R_1$, or in mathematical terms, the acceleration of the estimate (new cases or deaths) [8, 9]. Using this concepts and assumptions, we proposed a theoretical framework to classify the epidemic phase (Fig 1) in a specific region and time as following: i) Phase A—both high change of events and acceleration; ii) Phase B—high change of events but low acceleration; iii) Phase C—no

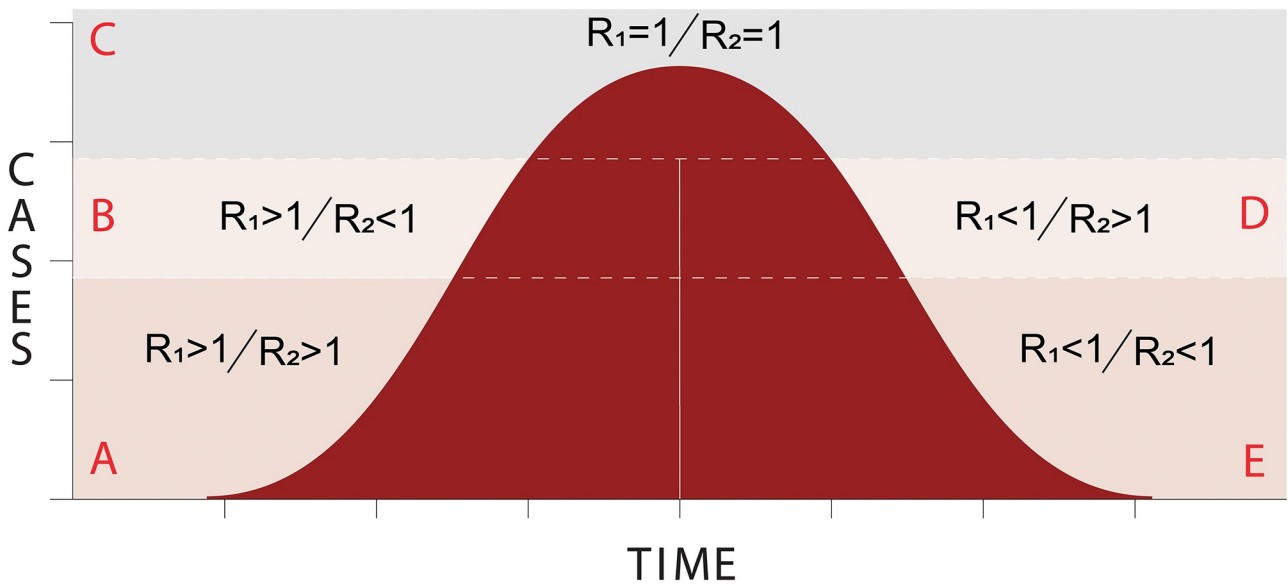

**Fig 1. Theoretical framework of epidemic phases based on Farr's law.**

change of events nor acceleration; iv) Phase D—small change of events but higher acceleration; and v) Phase E—both small change of events and acceleration.

Farr's law has not been widely used as other models using differential-equations were instead employed, such as the susceptible-infectious-removed (SIR) model [10]. The main reason is that it does not consider other important variables as population characteristics (immunity and susceptibility), public health interventions, and political actions against the pandemic. However, it may still be valuable and especially for new outbreaks where there is a lack of knowledge on parameters of disease such as Ebola [11], Chikungunya [11], opioid abuse [12], and indeed the current COVID-19 pandemic. In the study of 1990, Bregman et al. [9] showed a good prediction model for the cases of AIDS, showing that the peak was close, and it would happen a rapid deceleration which it did take place in the following years. Santillan et al. [11] compared the Farr's model with the incidence decay with exponential adjustment (IDEA) and SIR models. They reported Farr's Law mathematical approach to resemble solutions of SIR model in an asymptotic limit, where changes of transmission respond both to control policies and depletion of susceptible individuals. Moreover, they suggested the concept of the reproduction number ($R_0$) and the effects of control of epidemics (via behavior change or public health policy intervention) were implicitly captured in Farr's work (pre-microbial era).

In this study, we will model COVID-19 current data (until April 10, 2020) of new confirmed cases and deaths, from 210 countries as to test the assumptions of the 1840 Farr's law, to describe the epidemic dynamics, and also to make predictions to identify areas with high dynamic and suggest preparation and actions of health system in those regions.

## Methods

### Data

We extracted the COVID-19 data of total and new confirmed cases and deaths from all countries available in "Worldometer" website (210 countries, see Table 1) [13], until April 10, 2020. This website provides a real-time update on the actual number of COVID-19 cases worldwide, including the total confirmed cases, daily new confirmed cases, and severity of the disease (recovered, critical condition, or confirmed death) by country. Worldometer is composed of a team of researchers, developers, and volunteers with no political, governmental, or corporate affiliation to provide time relevant world statistics. As general information, it has been voted as one of the best free reference websites by the American Library Association, and its data has been used in the United Nations Conference Rio+20 [14].

In the context of COVID-19 pandemic, it has been a provider for important governmental institutions such as the UK, Thailand, Pakistan, Sri Lanka, and Vietnam Governments as well as for the COVID-19 dashboard by the Center for Systems Science and Engineering (CSSE) at John Hopkins University [15]. Based on its webpage, they obtain the data directly from official reports from the Government's communication channels or indirectly through local media sources when they are considered reliable.

We downloaded the data on new cases and deaths, separately, per day and country since the first available record (January 23, 2020). We extracted directly from the webpage code source from each plot using a standardized extraction spreadsheet to prevent losing data. Additionally, we extracted data of income, urban index, population density, population size, and proportion of the older population (>65 years old) from the World Bank data repository [16] of the included countries; we estimated the number of cases and deaths per 1000 and 100,000 inhabitants by country, respectively. Finally, we performed a post-hoc extraction of the new cases for June 10, 2020 (one point extraction), and the political actions against COVID-19 per country, from the University of Washington Health Index and Evaluation

**Table 1. Total cases, deaths, and Farr's ratios associated with COVID-19 pandemic, per country until April 10, 2020.**

| Region | Country | Income | Urban population (%) | Population Density Pop/Km² | Adults older than 65 y | Any Restriction Policy | First case reported* | Total Cases on April 10 | Cases/Population per 100,000 inhab. | Cases First ratio | Cases Percent increase | Cases Second ratio | Epidemic Phase | Predicted End Date | Epidemic Status June 10 | New Cases on June 10 | Total Deaths on April 10 | Mortality First ratio | Mortality Percent increase | Mortality Second ratio |
|---|---|---|---|---|---|---|---|---|---|---|---|---|---|---|---|---|---|---|---|---|
| North America | Canada | High | 81·41 | 4·08 | 17·23 | Yes | 21-Feb | 22,046 | 58·41 | 1·47 | 46·67% | 0·71 | B | 25-May | Descending | 472 | 556 | 2·71 | 171·49% | 1·46 |
| | Greenland | High | 86·82 | 0·14 | : | : | 18-Mar | 11 | 19·38 | 0·42 | -58·33% | 0·17 | E | 15-Abr | No cases* | 0 | : | : | : | : |
| | Mexico | Upper Middle | 80·16 | 64·91 | 7·22 | Yes | 29-Mar | 3,441 | 2·67 | 1·68 | 68·39% | 0·84 | B | >Jun-10 | Increasing | 4,199 | 194 | 3·20 | 219·84% | 1·03 |
| | USA | High | 82·26 | 35·77 | 15·81 | Yes | 21-Feb | 489,268 | 147·81 | 1·52 | 51·63% | 0·70 | B | 30-May | Descending | 20,852 | 18,015 | 2·73 | 172·59% | 0·78 |
| Central America | Belize | Upper Middle | 45·72 | 16·79 | 4·74 | : | 25-Mar | 10 | 2·51 | 2·38 | 137·50% | 2·09 | A | : | : | : | 2 | : | : | : |
| | Costa Rica | Upper Middle | 79·34 | 97·91 | 9·55 | : | 8-Mar | 539 | 10·58 | 0·97 | -2·94% | 0·88 | E | 25-May | Increasing (2nd peak) | 86 | 3 | 0·00 | -100·00% | : |
| | El Salvador | Lower Middle | 72·02 | 309·88 | 8·29 | : | 21-Mar | 117 | 1·80 | 1·65 | 64·71% | 1·04 | A | : | : | : | 6 | 0·75 | -25·00% | 0·00 |
| | Guatemala | Upper Middle | 51·05 | 160·95 | 4·81 | : | 16-Mar | 126 | 0·70 | 1·54 | 54·22% | 1·25 | A | : | : | : | 3 | 0·00 | -100·00% | : |
| | Honduras | Lower Middle | 57·10 | 85·69 | 4·69 | Yes | 14-Mar | 382 | 3·86 | 1·45 | 45·15% | 0·70 | B | 15-May | Increasing | 485 | 23 | 1·33 | 33·00% | 0·06 |
| | Nicaragua | Lower Middle | 58·52 | 53·73 | 5·25 | : | 20-Mar | 7 | 0·11 | 0·83 | -16·67% | 1·08 | D | . | : | : | 1 | : | : | : |
| | Panama | High | 67·71 | 56·19 | 8·10 | Yes | 11-Mar | 2,752 | 63·78 | 1·43 | 42·90% | 0·75 | B | 20-May | Increasing | 656 | 66 | 1·95 | 94·62% | 0·82 |
| South America | Argentina | Upper Middle | 91·87 | 16·26 | 11·12 | Yes | 5-Mar | 1,894 | 4·19 | 1·40 | 40·43% | 0·65 | B | 15-May | Increasing | 1,226 | 81 | 2·37 | 136·62% | 0·99 |
| | Bolivia | Lower Middle | 69·43 | 10·48 | 7·19 | Yes | 12-Mar | 268 | 2·30 | 2·56 | 156·16% | 2·10 | A | : | : | : | 19 | 0·86 | -14·29% | 0·49 |
| | Brazil | Upper Middle | 86·57 | 25·06 | 8·92 | Yes | 29-Feb | 18,397 | 8·65 | 1·65 | 65·24% | 0·85 | B | >Jun-10 | Plateau peak | 33,100 | 974 | 2·23 | 122·77% | 0·80 |
| | Chile | High | 87·56 | 25·19 | 11·53 | Yes | 4-Mar | 6,501 | 34·01 | 1·66 | 65·92% | 0·80 | B | 5-Jun | Increasing | 5,737 | 65 | 2·28 | 127·53% | 0·77 |
| | Colombia | Upper Middle | 80·78 | 44·75 | 8·48 | Yes | 9-Mar | 2,223 | 4·37 | 1·47 | 47·48% | 1·09 | A | : | : | : | 69 | 2·26 | 125·61% | 1·00 |
| | Ecuador | Upper Middle | 63·82 | 68·79 | 7·16 | Yes | 1-Mar | 7,161 | 40·59 | 1·66 | 65·66% | 1·82 | A | : | : | : | 297 | 1·69 | 68·98% | 0·72 |
| | Falkland Islands | : | : | 0·25 | : | : | 5-Apr | 5 | 143·68 | : | : | : | : | : | : | : | : | : | : | : |
| | French Guiana | : | : | 3·39 | : | : | 11-Mar | 83 | 27·79 | 1·69 | 68·57% | 1·81 | A | : | : | : | : | 0·00 | -100·00% | : |
| | Guyana | Upper Middle | 26·61 | 3·96 | 6·45 | : | 12-Mar | 37 | 4·70 | : | : | : | : | : | : | : | 6 | 0·67 | -33·33% | 0·00 |
| | Paraguay | Upper Middle | 61·59 | 17·51 | 6·43 | : | 10-Mar | 129 | 1·81 | 1·84 | 83·94% | 1·51 | A | : | : | : | 6 | 0·00 | -100·00% | 0·00 |
| | Peru | Upper Middle | 77·91 | 24·99 | 8·09 | Yes | 7-Mar | 5,897 | 17·88 | 2·76 | 175·64% | 1·77 | A | : | : | : | 169 | 2·54 | 154·29% | 0·83 |
| | Suriname | Upper Middle | 66·06 | 3·69 | 6·91 | : | 20-Mar | 10 | 1·70 | : | : | : | : | : | : | : | 1 | : | : | : |
| | Uruguay | High | 95·33 | 19·71 | 14·81 | : | 14-Mar | 473 | 13·62 | 0·91 | -9·30% | 0·87 | E | 20-May | Descending | 1 | 7 | 3·00 | 200·00% | 0·00 |
| | Venezuela | Upper Middle | 88·21 | 32·73 | 7·27 | : | 15-Mar | 171 | 0·60 | 0·75 | -25·29% | 0·84 | E | 5-May | Descend not clear | 106 | 9 | 0·83 | -16·67% | 0·13 |

(*Continued*)

**Table 1.** (Continued)

| Region | Country | Income | Urban population (%) | Population Density Pop/Km² | Adults older than 65 y | Any Restriction Policy | First case reported* | Total Cases on April 10 | Cases/Population per 100,000 inhab. | First ratio | Percent increase | Second ratio | Epidemic Phase | Predicted End Date | Epidemic Status June 10 | New Cases on June 10 | Total Deaths on April 10 | First ratio | Percent increase | Second ratio |
|---|---|---|---|---|---|---|---|---|---|---|---|---|---|---|---|---|---|---|---|---|
| | | | | | | | | | | | | | | | | | | Mortality | | |
| Caribbean | Anguilla | .. | .. | 164·84 | .. | .. | 2-Apr | 3 | 20·00 | .. | .. | .. | .. | .. | .. | .. | .. | .. | .. | .. |
| | Antigua and Barbuda | High | 24·60 | 218·83 | 8·80 | .. | 23-Mar | 19 | 19·40 | .. | .. | .. | .. | .. | .. | .. | 2 | .. | .. | .. |
| | Aruba | High | 43·41 | 588·03 | 13·55 | .. | 17-Mar | 86 | 80·55 | 1·32 | 31·72% | 1·18 | A | .. | .. | .. | .. | .. | .. | .. |
| | Bahamas | High | 83·03 | 38·53 | 7·26 | .. | 19-Mar | 41 | 10·43 | 1·22 | 22·47% | 0·73 | B | 5-May | Descending | 0 | 8 | .. | .. | .. |
| | Barbados | High | 31·15 | 666·61 | 15·80 | .. | 19-Mar | 66 | 22·97 | 0·95 | -5·35% | 0·89 | E | 15-May | Descending | 4 | 4 | 0·25 | -75·00% | 0·00 |
| | Bermuda | High | 100·00 | 1184·59 | .. | .. | 22-Mar | 48 | 77·07 | 0·80 | -20·00% | 1·16 | D | .. | .. | .. | 4 | 0·00 | -100·00% | .. |
| | British Virgin Islands | High | 47·72 | 198·68 | .. | .. | 31-Mar | 3 | 9·92 | .. | .. | .. | .. | .. | .. | .. | .. | .. | .. | .. |
| | Caribbean Netherlands | High | .. | .. | .. | .. | 2-Apr | 2 | 7·63 | .. | .. | .. | .. | .. | .. | .. | 1 | .. | .. | .. |
| | Cayman Islands | High | 100·00 | 267·39 | .. | .. | 19-Mar | 45 | 68·47 | 2·20 | 120·33% | 1·76 | A | .. | .. | .. | 1 | .. | .. | .. |
| | Cuba | Upper Middle | 77·04 | 109·00 | 15-19 | Yes | 13-Mar | 564 | 4·98 | 1·84 | 84·46% | 1·00 | A | .. | .. | .. | 15 | 1·75 | 75·00% | 1·33 |
| | Curaçao | High | 89·15 | 367·12 | 16·68 | .. | 14-Mar | 14 | 8·53 | 0·58 | -41·67% | 0·93 | E | 25-Abr | Descending | 1 | 1 | .. | .. | .. |
| | Dominica | Upper Middle | 70·48 | 95·50 | .. | .. | 23-Mar | 16 | 22·23 | 1·50 | 50·00% | 10·25 | A | .. | .. | .. | .. | .. | .. | .. |
| | Dominican Republic | Upper Middle | 81·07 | 219·98 | 7·08 | Yes | 6-Mar | 2,620 | 24·15 | 1·37 | 36·93% | 0·54 | B | 5-May | Plateau peak | 393 | 126 | 3·50 | 250·27% | 1·93 |
| | Grenada | Upper Middle | 36·27 | 327·81 | 9·62 | .. | 26-Mar | 12 | 10·66 | .. | .. | .. | .. | .. | .. | .. | .. | .. | .. | .. |
| | Guadeloupe | .. | 94·78 | 245·70 | .. | .. | 14-Mar | 143 | 35·74 | 1·34 | 34·05% | 3·46 | A | .. | .. | .. | 8 | 0·44 | -55·56% | 0·17 |
| | Haiti | Low | 55·28 | 403·60 | 4·95 | .. | 23-Mar | 30 | 0·26 | 1·26 | 26·19% | 1·34 | A | .. | .. | .. | 2 | .. | .. | .. |
| | Jamaica | Upper Middle | 55·67 | 270·99 | 8·80 | .. | 11-Mar | 63 | 2·13 | 1·53 | 53·32% | 1·35 | A | .. | .. | .. | 4 | 0·25 | -75·00% | 0·00 |
| | Martinique | .. | .. | 333·33 | .. | .. | 10-Mar | 154 | 41·04 | 0·55 | -44·99% | 0·60 | E | 25-Abr | Descending | 0 | 6 | 0·75 | -25·00% | 0·00 |
| | Montserrat | .. | .. | 49·02 | .. | .. | 26-Mar | 9 | 180·29 | .. | .. | .. | .. | .. | .. | .. | .. | .. | .. | .. |
| | Saint Kitts and Nevis | High | 30·78 | 201·70 | .. | .. | 30-Mar | 11 | 20·68 | .. | .. | .. | .. | .. | .. | .. | .. | 0·29 | -71·43% | 0·88 |
| | Saint Lucia | High | 18·68 | 298·18 | 9·81 | .. | 15-Mar | 14 | 7·62 | 0·50 | -50·00% | 5·00 | D | .. | .. | .. | .. | .. | .. | .. |
| | Saint Martin | High | .. | 698·11 | .. | .. | 18-Mar | 32 | 82·76 | 0·92 | -7·78% | 0·50 | E | 25-Abr | No cases* | 0 | 2 | .. | .. | .. |
| | Saint Pierre Miquelon | .. | .. | 24·79 | .. | .. | 5-Apr | 1 | 17·26 | .. | .. | .. | .. | .. | .. | .. | .. | .. | .. | .. |
| | Sint Maarten | High | 100·00 | 1235·29 | .. | .. | 23-Mar | 50 | 116·62 | 3·82 | 282·46% | 2·21 | A | .. | .. | .. | 8 | .. | .. | .. |
| | St. Barth | High | .. | 476·19 | .. | .. | 16-Mar | 6 | 60·75 | .. | .. | .. | .. | .. | .. | .. | .. | .. | .. | .. |
| | St. Vincent Grenadines | High | 52·20 | 282·59 | 9·59 | .. | 1-Apr | 12 | 10·82 | .. | .. | .. | .. | .. | .. | .. | .. | .. | .. | .. |
| | Trinidad and Tobago | High | 53·18 | 270·93 | 10·73 | .. | 14-Mar | 109 | 7·79 | 0·93 | -7·22% | 2·13 | D | .. | .. | .. | 8 | 0·72 | -27·78% | 0·22 |
| | Turks and Caicos | High | 93·10 | 39·65 | .. | .. | 26-Mar | 8 | 20·66 | .. | .. | .. | .. | .. | .. | .. | 1 | .. | .. | .. |

*(Continued)*

**Table 1.** (Continued)

| Region | Country | Income | Urban population (%) | Population Density Pop/Km² | Adults older than 65 y | Any Restriction Policy | First case reported* | Cases | | | | | | | | | Mortality | | | |
|---|---|---|---|---|---|---|---|---|---|---|---|---|---|---|---|---|---|---|---|---|
| | | | | | | | | Total Cases on April 10 | Cases/Population per 100,000 inhab. | First ratio | Percent increase | Second ratio | Epidemic Phase | Predicted End Date | Epidemic Status June 10 | New Cases on June 10 | Total Deaths on April 10 | First ratio | Percent increase | Second ratio |
| Africa | Algeria | Upper Middle | 72·63 | 17·73 | 6·36 | : | 1-Mar | 1,761 | 4·02 | 1·25 | 25·03% | 0·96 | B | >Jun-10 | Descending | 102 | 256 | 3·10 | 209·70% | 2·00 |
| | Angola | Lower Middle | 65·51 | 24·71 | 2·22 | : | 21-mAR | 19 | 0·06 | 5·63 | 462·50% | 22·06 | A | : | : | : | 2 | : | : | : |
| | Benin | Low | 47·31 | 101·85 | 3·25 | : | 18-Mar | 35 | 0·29 | 2·03 | 102·78% | 2·07 | A | : | : | : | 1 | : | : | : |
| | Botswana | Upper Middle | 69·45 | 3·98 | 4·22 | : | 31-Mar | 13 | 0·55 | : | : | : | : | : | : | : | 1 | : | : | : |
| | Burkina Faso | Low | 29·36 | 72·19 | 2·41 | : | 10-Mar | 443 | 2·12 | 1·15 | 14·51% | 0·87 | B | 30-May | Descending | 0 | 24 | 5·00 | 400·00% | 1·84 |
| | Burundi | Low | 13·03 | 435·18 | 2·25 | : | 2-Apr | 3 | 0·03 | : | : | : | : | : | : | : | : | : | : | : |
| | Cabo Verde | Lower Middle | 65·73 | 134·93 | 4·61 | : | 21-Mar | 7 | 1·26 | : | : | : | : | : | : | : | 1 | : | : | : |
| | Cameroon | Lower Middle | 56·37 | 53·34 | 2·73 | : | 14-Mar | 803 | 3·02 | 2·49 | 149·24% | 1·06 | A | : | : | : | 12 | 0·60 | : | : |
| | CAR | Low | 41·36 | 7·49 | 2·83 | : | 20-Mar | 8 | 0·17 | : | : | : | : | : | : | : | : | : | : | : |
| | Chad | Low | 23·06 | 12·29 | 2·48 | : | 23-Mar | 11 | 0·07 | : | : | : | : | : | : | : | : | : | : | : |
| | Congo | Lower Middle | 66·92 | 15·36 | 2·68 | : | 19-Mar | 60 | 1·09 | 6·64 | 564·33% | 12·19 | A | : | : | : | 5 | 0·00 | -100·00% | . |
| | Djibouti | Low | 77·78 | 41·37 | 4·53 | : | 23-Mar | 150 | 15·18 | 2·33 | 133·21% | 1·43 | A | : | : | : | 1 | : | : | : |
| | DRC | Upper Middle | 44·46 | 37·08 | 3·02 | : | 13-Mar | 215 | 0·24 | 1·37 | 36·74% | 0·98 | B | >Jun-10 | Descend not clear | 131 | 20 | 0·73 | -26·67% | 0·30 |
| | Egypt | Lower Middle | 42·70 | 98·87 | 5·23 | Yes | 1-Mar | 1,699 | 1·66 | 1·59 | 58·78% | 1·18 | A | : | : | : | 118 | 1·79 | 78·64% | 0·92 |
| | Equatorial Guinea | Upper Middle | 72·14 | 46·67 | 2·46 | : | 18-Mar | 18 | 1·28 | 0·78 | -22·22% | 2·22 | D | : | : | : | : | : | : | : |
| | Eritrea | Low | : | 29·36 | : | : | 25-Mar | 34 | 0·96 | 0·45 | -55·21% | 0·38 | E | : | : | : | : | : | : | : |
| | Eswatini | Lower Middle | 23·80 | 66·06 | 4·01 | : | 22-Mar | 12 | 1·03 | : | : | : | : | : | : | : | : | : | : | : |
| | Ethiopia | Low | 20·76 | 109·22 | 3·50 | : | 15-Mar | 65 | 0·06 | 1·79 | 78·97% | 1·43 | A | : | : | : | 3 | : | : | : |
| | Gabon | Upper Middle | 89·37 | 8·22 | 3·56 | : | 17-Mar | 44 | 1·98 | 5·38 | 438·10% | 9·62 | A | : | : | : | 1 | : | : | : |
| | Gambia | Low | 61·27 | 225·31 | 2·59 | : | 23-Mar | 4 | 0·17 | : | : | : | . | : | : | : | 1 | : | : | : |
| | Ghana | Lower Middle | 56·06 | 130·82 | 3·07 | : | 15-Mar | 378 | 1·22 | 2·52 | 152·34% | 1·30 | A | : | : | : | 6 | : | : | : |
| | Guinea | Low | 36·14 | 50·52 | 2·93 | : | 20-Mar | 194 | 1·48 | 4·61 | 361·00% | 2·93 | A | : | : | : | : | : | : | : |
| | Guinea-Bissau | Low | 43·36 | 66·65 | 2·82 | : | 30-Mar | 36 | 1·83 | : | : | : | : | : | : | : | : | : | : | : |
| | Ivory Coast | Lower Middle | 50·78 | 78·83 | 2·86 | : | 14-Mar | 444 | 1·68 | 3·74 | 273·91% | 1·40 | A | : | : | : | 3 | : | : | : |
| | Kenya | Lower Middle | 27·03 | 90·30 | 2·34 | : | 15-Mar | 189 | 0·35 | 1·47 | 46·93% | 0·64 | B | 10-May | Descend not clear | 105 | 7 | 0·50 | -50·00% | 0·00 |
| | Liberia | Low | 51·15 | 50·03 | 3·25 | : | 17-Mar | 37 | 0·73 | : | : | : | : | : | : | : | 5 | 0·33 | : | 0·00 |
| | Libya | Upper Middle | 80·10 | 3·80 | 4·39 | : | 28-Mar | 24 | 0·35 | : | : | : | : | : | : | : | 1 | : | : | : |
| | Madagascar | Low | 37·19 | 45·14 | 2·99 | : | 23-Mar | 93 | 0·34 | 0·92 | -7·94% | 1·72 | D | : | : | : | : | : | : | : |
| | Malawi | Low | 16·94 | 192·44 | 2·65 | : | 4-Apr | 9 | 0·05 | : | : | : | : | : | : | : | 1 | : | : | : |
| | Mali | Low | 42·36 | 15·64 | 2·51 | : | 26-Mar | 87 | 0·43 | 1·18 | 17·62% | 1·10 | A | : | : | : | 7 | 0·67 | -33·33% | 0·50 |
| | Mauritania | Lower Middle | 53·67 | 4·27 | 3·14 | : | 18-Mar | 7 | 0·15 | : | : | : | : | : | : | : | 1 | : | : | : |

(Continued)

Table 1. (Continued)

| Region | Country | Income | Urban population (%) | Population Density Pop/Km² | Adults older than 65 y | Any Restriction Policy | First case reported* | Cases | | | | | | | | | | Mortality | | |
|---|---|---|---|---|---|---|---|---|---|---|---|---|---|---|---|---|---|---|---|---|
| | | | | | | | | Total Cases on April 10 | Cases/Population per 100,000 inhab. | First ratio | Percent increase | Second ratio | Epidemic Phase | Predicted End Date | Epidemic Status June 10 | New Cases on June 10 | Total Deaths on April 10 | First ratio | Percent increase | Second ratio |
| | Mauritius | Upper Middle | 40·79 | 623·30 | 11·47 | :: | 19-Mar | 318 | 25·00 | 1·05 | 4·85% | 0·77 | B | :: | :: | :: | 9 | 1·67 | 66·67% | 2·00 |
| | Mayotte | :: | :: | 695·19 | :: | :: | 16-Mar | 191 | 70·01 | 1·37 | 37·09% | 1·10 | A | :: | :: | :: | 2 | :: | :: | :: |
| | Morocco | Lower Middle | 62·45 | 80·73 | 7·01 | :: | 5-Mar | 1,448 | 3·92 | 1·88 | 88·24% | 1·09 | A | :: | :: | :: | 107 | 4·61 | 360·86% | 2·67 |
| | Mozambique | Low | 35·99 | 37·51 | 2·89 | :: | 24-Mar | 20 | 0·06 | :: | :: | :: | :: | :: | :: | :: | :: | :: | :: | :: |
| | Namibia | Upper Middle | 50·03 | 2·97 | 3·64 | :: | 19-Mar | 16 | 0·63 | 0·67 | -33·33% | 1·00 | D | :: | :: | :: | :: | :: | :: | :: |
| | Niger | Low | 16·43 | 17·72 | 2·60 | :: | 22-Mar | 410 | 1·69 | 4·06 | 306·18% | 1·07 | A | :: | :: | :: | 11 | 1·50 | 50·00% | 0·00 |
| | Nigeria | Lower Middle | 50·34 | 215·06 | 2·75 | :: | 28-Feb | 288 | 0·14 | 1·42 | 42·08% | 1·23 | A | :: | :: | :: | 7 | 1·22 | 22·22% | 1·61 |
| | Réunion | :: | 99·14 | 351·65 | :: | :: | 12-Mar | 382 | 42·67 | 0·65 | -35·26% | 1·23 | D | :: | :: | :: | :: | :: | :: | :: |
| | Rwanda | Low | 17·21 | 498·66 | 2·94 | :: | 15-Mar | 113 | 0·87 | 0·95 | -5·50% | 0·77 | E | 5-May | Increasing | 41 | :: | :: | :: | :: |
| | Sao Tome and Principe | Lower Middle | 72·80 | 219·82 | 2·93 | :: | 6-Mar | 4 | 1·83 | :: | :: | :: | :: | :: | :: | :: | 1 | :: | :: | :: |
| | Senegal | Lower Middle | 47·19 | 82·35 | 3·09 | :: | 3-Mar | 265 | 1·58 | 1·00 | 0·50% | 0·89 | B | 25-May | Plateau peak | 124 | 2 | 0·00 | -100·00% | :: |
| | Seychelles | High | 56·69 | 210·35 | 7·59 | :: | 15-Mar | 11 | 11·18 | 1·17 | 16·67% | 2·08 | A | :: | :: | :: | :: | :: | :: | :: |
| | Sierra Leone | Low | 42·06 | 105·99 | 2·97 | :: | 1-Apr | 8 | 0·10 | :: | :: | :: | A | :: | :: | :: | :: | :: | :: | :: |
| | Somalia | Low | 44·97 | 23·92 | 2·87 | :: | 26-Mar | 21 | 0·13 | 1·53 | 52·50% | 1·03 | A | :: | :: | :: | 1 | :: | :: | :: |
| | South Africa | Upper Middle | 66·36 | 47·63 | 5·32 | :: | 7-Mar | 2,003 | 3·38 | 0·99 | -0·53% | 1·30 | D | :: | :: | :: | 24 | 1·75 | :: | 0·49 |
| | South Sudan | Low | 19·62 | 17·71 | 3·40 | :: | 7-Apr | 4 | 0·04 | 1·00 | :: | :: | :: | :: | :: | :: | :: | :: | :: | :: |
| | Sudan | Lower Middle | 34·64 | 22·16 | 3·58 | :: | 18-Mar | 15 | 0·03 | 2·08 | 108·33% | 1·70 | A | :: | :: | :: | 2 | :: | :: | :: |
| | Tanzania | Low | 33·78 | 63·58 | 2·60 | :: | 18-Mar | 32 | 0·05 | 1·28 | 27·78% | 2·42 | A | :: | :: | :: | 3 | :: | :: | :: |
| | Togo | Low | 41·70 | 145·05 | 2·87 | :: | 20-Mar | 76 | 0·92 | 1·93 | 92·86% | 2·89 | A | :: | :: | :: | 3 | 0·00 | -100·00% | :: |
| | Tunisia | Lower Middle | 68·95 | 74·44 | 8·32 | :: | 8-Mar | 671 | 5·68 | 2·05 | 105·24% | 0·94 | B | :: | :: | :: | 25 | 1·53 | 52·78% | 1·27 |
| | Uganda | Low | 23·77 | 213·06 | 1·94 | :: | 23-Mar | 53 | 0·12 | 0·20 | -80·42% | 0·29 | E | :: | :: | :: | :: | 0·13 | -86·94% | 0·19 |
| | Western Sahara | :: | :: | 2·13 | :: | :: | 4-Apr | 4 | 0·67 | :: | :: | :: | :: | :: | :: | :: | :: | :: | :: | :: |
| | Zambia | Lower Middle | 43·52 | 23·34 | 2·10 | :: | 22-Mar | 40 | 0·22 | :: | :: | :: | :: | :: | :: | :: | 2 | :: | :: | :: |
| | Zimbabwe | Lower Middle | 32·21 | 37·32 | 2·94 | :: | 21-Mar | 11 | 0·07 | 0·83 | -16·67% | 1·08 | D | :: | :: | :: | 3 | :: | :: | :: |
| Asia | Afghanistan | Low | 25·50 | 56·94 | 2·58 | :: | 7-Mar | 521 | 1·34 | 1·44 | 43·62% | 0·70 | B | 15-May | Descend not clear | 683 | 15 | 2·00 | 100·00% | 1·50 |
| | Armenia | Upper Middle | 63·15 | 103·68 | 11·25 | :: | 12-Mar | 937 | 31·62 | 1·18 | 17·79% | 0·86 | B | 30-May | Descend not clear | 428 | 12 | 0·92 | -8·33% | 1·69 |
| | Azerbaijan | Upper Middle | 55·68 | 120·27 | 6·20 | :: | 29-Feb | 991 | 9·77 | 2·11 | 111·02% | 0·96 | B | :: | :: | :: | 10 | 2·17 | 116·67% | 4·13 |
| | Bahrain | High | 89·29 | 2017·27 | 2·43 | :: | 25-Feb | 913 | 53·66 | 1·15 | 14·55% | 0·94 | B | >Jun-10 | Descend not clear | 469 | 6 | 0·83 | -16·67% | 0·13 |
| | Bangladesh | Lower Middle | 36·63 | 1239·58 | 5·16 | :: | 14-Mar | 424 | 0·26 | 4·34 | 334·03% | 13·66 | A | :: | :: | :: | 27 | 1·67 | 66·67% | 8·50 |
| | Bhutan | Lower Middle | 40·90 | 19·78 | 6·00 | :: | 20-Mar | 5 | 0·65 | :: | :: | :: | :: | :: | :: | :: | :: | :: | :: | :: |
| | Brunei | High | 77·63 | 81·40 | 4·87 | :: | 10-Mar | 136 | 31·09 | 0·37 | -63·26% | 1·32 | D | :: | :: | :: | 1 | :: | :: | :: |

(Continued)

**Table 1.** (Continued)

| Region | Country | Income | Urban population (%) | Population Density Pop/Km² | Adults older than 65 y | Any Restriction Policy | First case reported* | Cases Total Cases on April 10 | Cases/Population per 100,000 inhab. | First ratio | Percent increase | Second ratio | Epidemic Phase | Predicted End Date | Epidemic Status June 10 | New Cases on June 10 | Mortality Total Deaths on April 10 | First ratio | Percent increase | Second ratio |
|---|---|---|---|---|---|---|---|---|---|---|---|---|---|---|---|---|---|---|---|---|
| | Cambodia | Lower Middle | 23·39 | 92·06 | 4·57 | :· | 7-Mar | 119 | 0·71 | 0·68 | -32·18% | 1·03 | D | :· | :· | :· | :· | 0·00 | -100·00% | :· |
| | China | Upper Middle | 59·15 | 148·35 | 10·92 | :· | Before 23-Feb | 81,907 | 5·69 | 1·02 | 1·96% | 1·21 | A | :· | :· | :· | 3,336 | 0·67 | -32·79% | 0·88 |
| | Cyprus | High | 66·81 | 128·71 | 13·72 | Yes | 11-Mar | 595 | 49·28 | 1·44 | 43·85% | 0·92 | B | > Jun-10 | Descending | 2 | 10 | 0·33 | -66·67% | 0·00 |
| | Georgia | Upper Middle | 58·63 | 65·28 | 14·87 | :· | 28-Feb | 234 | 5·87 | 1·57 | 57·02% | 1·08 | A | :· | :· | :· | 3 | :· | :· | :· |
| | Hong Kong | High | 100·00 | 7096·19 | 16·88 | :· | 16-Feb | 990 | 13·21 | 1·01 | 1·29% | 0·85 | B | 20-May | Descending | 0 | 4 | :· | :· | :· |
| | India | Lower Middle | 34·03 | 454·94 | 6·18 | :· | 2-Mar | 7,598 | 0·55 | 2·24 | 124·42% | 1·18 | A | :· | :· | :· | 246 | 2·71 | 170·59% | 0·99 |
| | Indonesia | Upper Middle | 55·33 | 147·75 | 5·86 | :· | 6-Mar | 3,512 | 1·28 | 1·38 | 38·10% | 0·94 | B | > Jun-10 | :· | :· | 306 | 1·33 | 32·76% | 0·89 |
| | Iran | Upper Middle | 74·90 | 50·22 | 6·18 | :· | 21-Feb | 68,192 | 81·19 | 1·33 | 32·54% | 1·03 | A | :· | :· | :· | 4,232 | 1·01 | 0·82% | 1·02 |
| | Iraq | Upper Middle | 70·47 | 88·53 | 3·32 | :· | 25-Mar | 1,279 | 3·18 | 1·48 | 48·13% | 1·02 | A | :· | :· | :· | 70 | 1·27 | 27·41% | 0·89 |
| | Israel | High | 92·42 | 410·53 | 11·98 | Yes | 23-Feb | 10,095 | 116·63 | 1·72 | 71·74% | 0·67 | B | 20-May | Increasing (2nd peak) | 175 | 95 | 4·80 | 379·67% | 0·38 |
| | Japan | High | 91·62 | 347·07 | 27·58 | Yes | 16-Feb | 5,530 | 4·37 | 1·95 | 95·03% | 1·04 | A | :· | :· | :· | 99 | 1·31 | 30·61% | 1·31 |
| | Jordan | Upper Middle | 90·98 | 112·14 | 3·85 | :· | 15-Mar | 372 | 3·65 | 1·05 | 4·68% | 1·28 | A | :· | :· | :· | 7 | :· | :· | :· |
| | Kazakhstan | Upper Middle | 57·43 | 6·77 | 7·39 | Yes | 14-Mar | 812 | 4·32 | 2·37 | 137·17% | 1·61 | A | :· | :· | :· | 10 | 2·00 | 100·00% | 0·69 |
| | Kuwait | High | 100·00 | 232·17 | 2·55 | :· | 25-Feb | 993 | 23·25 | 2·24 | 124·11% | 1·43 | A | :· | :· | :· | 1 | :· | :· | :· |
| | Kyrgyzstan | Lower Middle | 36·35 | 32·93 | 4·49 | :· | 20-Mar | 298 | 4·57 | 1·66 | 65·60% | 0·91 | B | :· | :· | :· | 5 | :· | :· | :· |
| | Laos | Lower Middle | 35·00 | 30·60 | 4·08 | :· | 25-Mar | 16 | 0·22 | 1·25 | 25·00% | 3·13 | A | :· | :· | :· | :· | :· | :· | :· |
| | Lebanon | Upper Middle | 88·59 | 669·49 | 7·00 | :· | 26-Feb | 609 | 8·92 | 0·74 | -26·10% | 0·83 | E | 10-May | Descend not clear | 20 | 20 | 0·79 | -20·83% | 0·19 |
| | Macao | High | 100·00 | 20777·50 | 10·48 | :· | 15-Mar | 45 | 6·93 | 0·51 | -48·68% | 0·67 | E | 20-Abr | No cases* | 0 | :· | :· | :· | · |
| | Malaysia | Upper Middle | 76·04 | 95·96 | 6·67 | Yes | 28-Feb | 4,346 | 13·43 | 1·14 | 13·64% | 1·03 | A | :· | :· | :· | 70 | 0·89 | -10·78% | 0·59 |
| | Maldives | Upper Middle | 39·81 | 1718·99 | 3·70 | :· | 8-Mar | 19 | 3·51 | :· | :· | :· | :· | :· | :· | :· | :· | :· | :· | :· |
| | Mongolia | Lower Middle | 68·45 | 2·04 | 4·08 | :· | 16-Mar | 16 | 0·49 | 0·83 | -16·67% | 1·21 | D | :· | :· | :· | 1 | :· | :· | :· |
| | Myanmar | Lower Middle | 30·58 | 82·24 | 5·78 | :· | 24-Mar | 27 | 0·05 | 0·79 | -20·83% | 0·84 | E | :· | :· | :· | 3 | :· | :· | :· |
| | Nepal | Low | 19·74 | 195·94 | 5·73 | :· | 23-Mar | 9 | 0·03 | :· | :· | :· | :· | :· | :· | :· | :· | :· | :· | :· |
| | Oman | High | 84·54 | 15·60 | 2·39 | :· | 25-Mar | 484 | 9·48 | 1·76 | 76·48% | 1·39 | A | :· | :· | :· | 3 | 1·00 | 0·00% | 1·00 |
| | Pakistan | Lower Middle | 36·67 | 275·29 | 4·31 | :· | 29-Feb | 4,695 | 2·13 | 1·44 | 44·27% | 1·02 | A | :· | :· | :· | 66 | 1·66 | 65·78% | 0·86 |
| | Palestine | Lower Middle | 76·16 | 758·98 | :· | :· | 6-Mar | 267 | 5·23 | 2·11 | 110·52% | 1·18 | A | :· | :· | :· | 2 | · | :· | · |
| | Philippines | Lower Middle | 46·91 | 357·69 | 5·12 | Yes | 6-Mar | 4,195 | 3·83 | 0·92 | -7·95% | 0·57 | E | 5-May | Descend not clear | 740 | 221 | 1·50 | 50·44% | 0·76 |
| | Qatar | High | 93·58 | 239·59 | 1·37 | :· | 1-Mar | 2,512 | 87·19 | 2·76 | 176·45% | 2·06 | A | :· | :· | :· | 6 | 1·00 | 0·00% | 1·00 |
| | South Korea | High | 81·46 | 529·65 | 3·31 | Yes | 16-Feb | 10,450 | 20·38 | 0·85 | -14·52% | 0·85 | E | 20-May | Descending | 50 | 208 | 0·99 | -0·91% | 1·02 |
| | Saudi Arabia | High | 83·84 | 15·68 | 11·46 | :· | 4-Mar | 3,651 | 10·49 | 1·41 | 40·65% | 0·87 | B | > Jun-10 | Increasing | 3,717 | 47 | 3·01 | 201·28% | 0·32 |

*(Continued)*

**Table 1.** (Continued)

| Region | Country | Income | Urban population (%) | Population Density Pop/Km² | Adults older than 65 y | Any Restriction Policy | First case reported* | Cases | | | | | | | | Mortality | | | |
|---|---|---|---|---|---|---|---|---|---|---|---|---|---|---|---|---|---|---|---|
| | | | | | | | | Total Cases on April 10 | Cases/Population per 100,000 inhab. | First ratio | Percent increase | Second ratio | Epidemic Phase | Predicted End Date | Epidemic Status June 10 | New Cases on June 10 | Total Deaths on April 10 | First ratio | Percent increase | Second ratio |
| | Singapore | High | 100·00 | 7953·00 | 14·42 | : | 16-Feb | 2,108 | 36·03 | 1·53 | 52·80% | 1·25 | A | : | : | : | 7 | 1·00 | 0·00% | 2·13 |
| | Sri Lanka | Upper Middle | 18·48 | 345·56 | 10·47 | : | 11-Mar | 190 | 0·89 | 1·29 | 29·05% | 2·84 | A | : | : | : | 7 | 0·72 | -27·78% | 0·22 |
| | Syria | Low | 54·16 | 92·07 | 4·50 | : | 25-Mar | 19 | 0·11 | : | : | : | : | : | : | : | 2 | : | : | : |
| | Taiwan | High | 27·13 | 655·54 | : | : | 16-Feb | 382 | 1·60 | 0·64 | -36·32% | 0·75 | E | 30-Abr | Descending | 0 | 6 | : | : | : |
| | Thailand | Upper Middle | 49·95 | 135·90 | 11·90 | : | 17-Feb | 2,473 | 3·54 | 1·40 | 40·40% | 0·70 | B | 15-May | Descending | 4 | 33 | 1·97 | 97·22% | 2·11 |
| | Timor-Leste | Lower Middle | 30·58 | 85·27 | 4·32 | : | 10-Apr | 2 | 0·15 | : | : | : | : | : | : | : | : | : | : | : |
| | Turkey | Upper Middle | 75·14 | 106·96 | 8·48 | Yes | 13-Mar | 47,029 | 55·76 | 2·18 | 117·79% | 0·90 | B | : | : | : | 1,006 | 2·12 | 112·13% | 0·81 |
| | UAE | High | 86·52 | 135·61 | 1·09 | : | 16-Mar | 3,360 | 33·97 | 2·11 | 110·58% | 1·02 | A | : | : | : | 14 | 0·33 | -66·67% | 0·50 |
| | Uzbekistan | Lower Middle | 50·48 | 77·47 | 4·42 | : | 16-Mar | 624 | 1·86 | 2·73 | 173·19% | 2·35 | A | : | : | : | 3 | : | : | : |
| | Vietnam | Lower Middle | 35·92 | 308·13 | 7·27 | : | 6-Mar | 257 | 0·26 | 0·51 | -49·44% | 0·95 | E | 30-Abr | Descending | 0 | : | 1·61 | 60·52% | 0·65 |
| | Yemen | Low | 36·64 | 53·98 | 2·88 | : | 10-Apr | 1 | 0·00 | : | : | : | : | : | : | : | : | : | : | : |
| Europe | Albania | Upper Middle | 60·32 | 104·61 | 13·74 | : | 9-Mar | 416 | 14·46 | 0·90 | -10·02% | 0·87 | E | 20-May | Increasing (2nd peak) | 42 | 23 | 6·00 | 500·00% | 0·63 |
| | Andorra | High | 88·06 | 163·84 | : | : | 15-Mar | 601 | 777·84 | 1·28 | 28·27% | 1·05 | A | : | : | : | 26 | 3·06 | 205·82% | 0·44 |
| | Austria | High | 58·30 | 107·21 | 19·00 | Yes | 27-Feb | 13,551 | 150·46 | 1·13 | 13·05% | 0·63 | B | 10-May | Descending | 26 | 319 | 1·83 | 83·01% | 0·77 |
| | Belarus | Upper Middle | 78·60 | 46·73 | 14·85 | : | 3-Mar | 1,981 | 20·96 | 4·35 | 335·40% | 9·17 | A | : | : | : | 19 | 2·25 | : | 0·30 |
| | Belgium | High | 98·00 | 377·21 | 18·79 | Yes | 1-Mar | 26,667 | 230·09 | 1·56 | 55·86% | 0·92 | B | : | : | : | 3,019 | 2·30 | 130·14% | 0·65 |
| | Bosnia and Herzegovina | Upper Middle | 48·25 | 64·92 | 16·47 | : | 6-Mar | 901 | 27·46 | 1·34 | 34·03% | 0·75 | B | 20-May | Increasing (2nd peak) | 47 | 36 | 2·39 | 138·89% | 0·42 |
| | Bulgaria | Upper Middle | 75·01 | 64·70 | 21·02 | Yes | 8-Mar | 635 | 9·14 | 1·10 | 10·48% | 0·87 | B | 30-May | Increasing (2nd peak) | 104 | 25 | 1·71 | 71·43% | 0·15 |
| | Channel Islands | .. | 30·91 | 861·11 | 17·30 | : | 10-Mar | 398 | 228·92 | 1·72 | 71·88% | 0·66 | B | 15-May | Descending | 0 | 9 | 2·00 | 100·00% | 0·25 |
| | Croatia | Upper Middle | 56·95 | 73·08 | 20·45 | Yes | 26-Feb | 1,495 | 36·42 | 0·97 | -2·76% | 0·67 | E | 10-May | Descending | 2 | 21 | 2·28 | 127·78% | 1·49 |
| | Czechia | High | 73·79 | 137·60 | 19·42 | Yes | 2-Mar | 5,674 | 52·98 | 1·16 | 15·95% | 0·81 | B | 25-May | Descending | 73 | 119 | 2·14 | 114·11% | 0·72 |
| | Denmark | High | 87·87 | 138·07 | 19·81 | Yes | 28-Feb | 5,819 | 100·46 | 1·56 | 55·69% | 1·03 | A | : | : | : | 247 | 2·42 | 142·09% | 1·14 |
| | Estonia | High | 68·88 | 30·39 | 19·63 | Yes | 3-Mar | 1,258 | 94·83 | 1·63 | 62·99% | 2·20 | A | : | : | : | 24 | 5·50 | 450·00% | 0·20 |
| | Faeroe Islands | High | 42·06 | 34·74 | : | : | 6-Mar | 184 | 376·56 | 0·70 | -30·43% | 0·58 | E | 25-Abr | No cases* | 0 | : | : | : | : |
| | Finland | High | 85·38 | 18·16 | 21·72 | Yes | 26-Feb | 2,769 | 49·98 | 1·19 | 19·49% | 0·92 | B | >Ju-10 | Descending | 15 | 48 | 4·37 | 337·41% | 1·28 |
| | France | High | 80·44 | 122·34 | 20·03 | Yes | 25-Feb | 124,869 | 191·30 | 1·55 | 55·44% | 0·90 | B | 10-May | Descending | : | 13,197 | 1·88 | 87·54% | 0·83 |
| | Germany | High | 77·31 | 237·37 | 21·46 | Yes | 25-Feb | 119,624 | 142·78 | 1·14 | 13·75% | 0·80 | B | 30-May | Descending | 311^ | 2,607 | 2·52 | 151·72% | 0·79 |
| | Gibraltar | High | 100·00 | 3371·80 | : | : | 16-Mar | 127 | 376·96 | 3·81 | 281·11% | 6·48 | A | : | : | : | : | : | : | : |
| | Greece | High | 79·06 | 83·22 | 21·66 | Yes | 27-Feb | 2,011 | 19·29 | 1·40 | 39·90% | 1·44 | A | : | Descending | : | 91 | 1·26 | 26·01% | 0·80 |
| | Hungary | High | 71·35 | 107·91 | 19·16 | Yes | 5-Mar | 1,190 | 12·32 | 1·32 | 31·61% | 0·71 | B | 15-May | Descending | 10 | 77 | 2·12 | 112·06% | 4·28 |
| | Iceland | High | 93·81 | 3·53 | 14·80 | Yes | 1-Mar | 1,675 | 490·85 | 0·94 | -5·76% | 0·73 | E | 10-May | Descending | 0 | 6 | 1·75 | 75·24% | 1·01 |
| | Ireland | High | 63·17 | 70·45 | 13·87 | Yes | 3-Mar | 7,054 | 142·86 | 1·48 | 48·38% | 0·75 | B | 25-May | Descending | 16 | 287 | 0·67 | -33·13% | 1·15 |
| | Isle of Man | High | 52·59 | 147·50 | : | : | 20-Mar | 201 | 236·38 | 1·71 | 70·54% | 1·14 | A | : | : | : | 1 | : | : | : |
| | Italy | High | 70·44 | 205·45 | 22·75 | Yes | 21-Feb | 147,577 | 244·08 | 0·89 | -10·84% | 0·87 | E | 10-Jun | Descending | 202 | 18,849 | 0·96 | -4·21% | 0·79 |
| | Latvia | High | 68·14 | 30·98 | 20·04 | Yes | 8-Mar | 612 | 32·45 | 1·16 | 16·44% | 0·67 | B | 10-May | Descending | 3 | 3 | : | : | : |

*(Continued)*

**Table 1.** (Continued)

| Region | Country | Income | Urban population (%) | Population Density Pop/Km² | Adults older than 65 y | Any Restriction Policy | First case reported* | Cases | | | | | | | | | Mortality | | | |
|---|---|---|---|---|---|---|---|---|---|---|---|---|---|---|---|---|---|---|---|---|
| | | | | | | | | Total Cases on April 10 | Cases/Population per 100,000 inhab. | First ratio | Percent increase | Second ratio | Epidemic Phase | Predicted End Date | Epidemic Status June 10 | New Cases on June 10 | Total Deaths on April 10 | First ratio | Percent increase | Second ratio |
| | Liechtenstein | High | 14·34 | 236·94 | : | : | 11-Mar | 79 | 207·20 | 1·43 | 43·04% | 6·54 | A | : | : | : | 1 | : | : | : |
| | Lithuania | High | 67·68 | 44·53 | 19·71 | Yes | 28-Feb | 999 | 36·70 | 1·35 | 34·99% | 0·69 | B | 15-May | Descending | 6 | 22 | 3·11 | 211·11% | 4·53 |
| | Luxembourg | High | 90·98 | 250·09 | 14·18 | Yes | 5-Mar | 3,223 | 514·87 | 0·95 | -5·16% | 0·63 | E | 5-May | Descending | 3 | 54 | 5·02 | 402·38% | 32·70 |
| | Malta | High | 94·61 | 1511·03 | 20·35 | : | 9-Mar | 350 | 79·27 | 1·38 | 37·94% | 1·35 | A | : | : | : | 2 | : | : | : |
| | Moldova | Lower Middle | 42·63 | 123·52 | 11·47 | Yes | 10-Mar | 1,438 | 35·65 | 1·98 | 97·60% | 0·98 | B | : | : | : | 29 | 5·85 | 485·00% | : |
| | Monaco | High | 100·00 | 19306·93 | | : | 12-Mar | 90 | 229·35 | 0·97 | -3·22% | 0·86 | E | 15-May | Descending | 0 | 1 | 0·00 | -100·00% | : |
| | Montenegro | Upper Middle | 66·81 | 46·27 | 14·97 | : | 18-Mar | 255 | 40·60 | 1·26 | 26·31% | 1·28 | A | : | : | : | 2 | : | : | : |
| | Netherlands | High | 91·49 | 511·46 | 19·20 | Yes | 28-Feb | 23,097 | 134·80 | 1·31 | 31·16% | 0·77 | B | 30-May | Descending | 184 | 2,511 | 1·66 | 66·14% | 0·70 |
| | North Macedonia | Upper Middle | 57·96 | 82·59 | 13·67 | : | 6-Mar | 711 | 34·13 | 1·26 | 25·56% | 0·81 | B | 25-May | Descend not clear | 125 | 32 | 1·71 | 71·11% | 1·20 |
| | Norway | High | 82·25 | 14·55 | 17·05 | Yes | 27-Feb | 6,298 | 116·17 | 1·12 | 11·84% | 0·93 | B | >Jun-10 | Descending | 18 | 113 | 1·92 | 92·38% | 1·01 |
| | Poland | High | 60·06 | 124·04 | 17·52 | Yes | 6-Mar | 5,955 | 15·73 | 1·06 | 6·41% | 0·67 | B | 15-May | Descend not clear | 282 | 181 | 2·19 | 118·66% | 0·95 |
| | Portugal | High | 65·21 | 112·24 | 21·95 | Yes | 3-Mar | 15,472 | 151·74 | 1·27 | 26·65% | 0·80 | B | 30-May | Descending | 294 | 435 | 1·83 | 83·25% | 0·54 |
| | Romania | Upper Middle | 54·00 | 84·64 | 18·34 | Yes | 28-Feb | 5,467 | 28·42 | 2·23 | 123·46% | 1·42 | A | : | : | : | 270 | 1·92 | 92·34% | 0·67 |
| | Russia | Upper Middle | 74·43 | 8·82 | 14·67 | Yes | 2-Mar | 11,917 | 8·17 | 2·41 | 141·05% | 0·99 | B | : | : | : | 94 | 3·61 | 260·56% | 1·20 |
| | San Marino | High | 97·23 | 563·08 | : | : | 1-Mar | 344 | 1013·82 | 1·23 | 23·38% | 1·53 | A | : | : | : | 34 | 3·64 | 264·02% | 23·48 |
| | Serbia | Upper Middle | 56·09 | 79·83 | 18·35 | Yes | 9-Mar | 3,105 | 35·54 | 1·97 | 97·38% | 0·91 | B | : | : | : | 71 | 2·37 | 137·30% | 0·46 |
| | Slovakia | High | 53·73 | 113·29 | 15·63 | Yes | 7-Mar | 715 | 13·10 | 1·45 | 44·92% | 1·24 | A | : | : | : | 2 | : | : | : |
| | Slovenia | High | 54·54 | 102·64 | 19·61 | Yes | 5-Mar | 1,160 | 55·80 | 1·06 | 6·40% | 0·96 | B | >Jun-10 | Descending | 2 | 45 | 1·75 | 74·75% | 1·13 |
| | Spain | High | 80·32 | 93·53 | 19·38 | Yes | 24-Feb | 157,053 | 335·91 | 1·22 | 21·97% | 0·73 | B | 25-May | Descending | 314 | 15,970 | 1·07 | 7·21% | 0·68 |
| | Sweden | High | 87·43 | 25·00 | 20·10 | Yes | 26-Feb | 9,685 | 95·90 | 1·47 | 47·28% | 0·93 | B | : | : | : | 870 | 2·43 | 142·78% | 0·86 |
| | Switzerland | High | 73·80 | 215·52 | 18·62 | Yes | 1-Mar | 24,551 | 283·68 | 1·00 | 0·12% | 0·75 | B | 20-May | Descending | 23 | 1,001 | 1·53 | 53·06% | 0·70 |
| | UK | High | 83·40 | 274·83 | 18·40 | Yes | 23-Feb | 73,758 | 108·65 | 1·85 | 84·70% | 0·82 | B | >Jun-10 | Descending | 1,003 | 8,958 | 2·48 | 148·14% | 0·91 |
| | Ukraine | Lower Middle | 69·35 | 77·03 | 16·43 | Yes | 12-Mar | 2,203 | 5·04 | 2·18 | 118·00% | 0·81 | B | >Jun-10 | Plateau peak | 525 | 69 | 3·09 | 208·89% | 2·41 |
| | Vatican City | High | 100 | 2272·73 | : | : | 24-Mar | 8 | 998·75 | : | : | : | : | : | : | : | : | : | : | : |

(*Continued*)

**Table 1.** (Continued)

| Region | Country | Income | Urban population (%) | Population Density Pop/Km² | Adults older than 65 y | Any Restriction Policy | First case reported* | Total Cases on April 10 | Cases/Population per 100,000 inhab. | Cases | | | | | | | Total Deaths on April 10 | Mortality | | |
|---|---|---|---|---|---|---|---|---|---|---|---|---|---|---|---|---|---|---|---|---|
| | | | | | | | | | | First ratio | Percent increase | Second ratio | Epidemic Phase | Predicted End Date | Epidemic Status June 10 | New Cases on June 10 | | First ratio | Percent increase | Second ratio |
| Oceania | Australia | High | 86·01 | 3·25 | 15·66 | ·· | 20-Feb | 6,203 | 24·33 | 0·78 | -21·83% | 0·55 | E | 5-May | Descending | 9 | 53 | 4·07 | 307·41% | 12·59 |
| | Fiji | Upper Middle | 56·25 | 48·36 | 5·45 | ·· | 21-Mar | 16 | 1·78 | 3·00 | 200·00% | 4·00 | A | ·· | ·· | ·· | ·· | ·· | ·· | ·· |
| | French Polynesia | ·· | 61·83 | 75·87 | 8·29 | ·· | 13-Mar | 51 | 18·16 | 1·18 | 17·94% | 1·99 | A | ·· | ·· | ·· | ·· | ·· | ·· | ·· |
| | New Caledonia | ·· | 70·68 | 15·54 | 9·17 | ·· | 21-mar | 18 | 6·30 | 0·30 | -70·00% | 0·48 | E | ·· | ·· | ·· | ·· | ·· | ·· | ·· |
| | New Zealand | High | 86·54 | 18·55 | 15·65 | ·· | 3-Mar | 1,283 | 26·61 | 3·16 | 215·66% | 0·73 | B | 5-Jun | No cases* | 0 | 2 | ·· | ·· | ·· |
| | Papua New Guinea | Lower Middle | 13·17 | 19·00 | 3·45 | ·· | 6-Apr | 2 | 0·02 | · | ·· | ·· | ·· | ·· | ·· | ·· | ·· | ·· | ·· | ·· |

USA: United States of America; CAR: Central African Republic; DRC: Democratic Republic of the Congo; UAE: United Arab Emirates; UK: United Kingdom; inhab: inhabitants· (··): No data· >

June 10: the predicted end date goes beyond that date· * Approximate date of first case reported· A: First and Second ratio higher than 1·0; B: First ratio higher than 1·0 but Second ratio less than 1·0;

C: Both ratios are equal to 1; D: First ratio less than 1·0 but Second ratio higher than 1·0; E: both ratios are less than 1·0·

*No cases for more than 10 days··

^Data belong to June 09, on June 10 there was no cases reported··

The countries with end date beyond June 10 are countries having more than zero predicted cases or deaths after we ended our simulation (June 10, 2020), meaning they could continue with the pandemic wave beyond the end of our simulation.

Center (IHME) database [17]. We used this analysis to compare current data with our predictions on the COVID-19 worldwide trend, including the countries with reduced cases near zero, and the countries with high predicted dynamics. We exported the data to a standardized spreadsheet in Excel Microsoft 2019 to performed data cleaning.

## Calculation of ratios

According to Farr's law, the intrinsic epidemic's behavior could be described as the relation of two arithmetic ratios:

1. The first ratio (R1) represents the change of cases or deaths (first level dynamic) comparing one time against the immediately before time (could be in days or months, based on the natural history of the disease). Thus, by subtracting 1·0 from this ratio, we can calculate the percent increase of cases or deaths, in physical terms, this measure could be understood as the "velocity of spread" of the epidemic.

2. The second ratio (R2) measures the rate of change of R1s. It compares the R1 of one time against the R1 from the immediately before time. In physical terms, we can interpret this ratio as the acceleration of the epidemic (new cases or deaths).

We calculated the Farr's ratios based on prior reports [9, 11, 12], the first ratio ($R_1$) was the division of given cases or deaths (I) at $t$ time over that estimate of $t$ time before (formula 1). Then, dividing the first ratio of one time over the immediately before resulted in the second ratio ($R_2$)(formula 2).

$$R_1 = \left( \frac{I(t+1)}{I(t)} \right) \tag{1}$$

$$R_2 = \frac{\left( \frac{I(t+3)}{I(t+2)} \right)}{\left( \frac{I(t+1)}{I(t)} \right)} \tag{2}$$

We predefined the sum of events over five days of consecutive data as a time length to calculate the ratios, as was suggested in prior studies to use clustered data to stabilize the data distribution [11]. We constructed the epidemic curves of all the 210 countries using those clustered data on new cases and deaths. The normality of the data was examined using skewness and kurtosis without data transformations to assure the estimates' interpretability. The countries with disease data less than 15 days (to calculate at least one $R_2$) were excluded from the ratio's calculation analysis.

## Prediction models and statistical analysis

We fit a normal curve to these data to use Farr's law as a predictive model of epidemic dynamics. First, similar to previous studies [9, 11, 12], we assumed the future $R_1$ and $R_2$ values would be the same as the mean of the past three-time intervals (five days), then we predicted the future first ratio and daily confirmed new cases and deaths, for the next two months (until June 10, 2020), by back-calculation, using the mean of the last three calculated $R_2$ [9]. To fit a normal curve, we set two assumptions [9, 11]: i) the $R_2$ was constant, having a value between 0·0 and 1·0, which signifies a constant deceleration in the rate of change ($R_1$). ii) each included country should report data of three-time intervals at minimum. The countries which do not fulfill these criteria were excluded from the predictive analysis [11].

Additionally, we performed ecological correlation analyses to assess the relationship between $R_1$ and $R_2$ of new cases and deaths, with urban index, population density, cases/1000 inhabitants, and deaths/100,000 inhabitants. The predefined hypothesis was higher $R_1$, and $R_2$ ratios are correlated with higher numbers of cases [9, 11] adjusted by population size, and these ratios are related to the number of urban areas and population density in the country.

We calculated 95% confidence intervals using the standard error of the mean of the last three calculated $R_2$. Two-sided $p<0.05$ was considered statistically significant. The analyses were performed in Microsoft Excel 2019 and Stata 15 (StataCorp LLC: College Station, TX).

For the report of this study, we followed the Guidelines for Accurate and Transparent Health Estimates Reporting (GATHER) to define best reporting practices for studies that compute health estimates for multiple populations in different times and spaces [18]. The checklist is reported in S1 Table. This study is exempt from institutional review board's review due to the use of publicly available and de-identified information.

## Results

### Worldwide COVID-19 dynamics based on $R_1$ and $R_2$

We obtained information from 210 countries, including reported new confirmed cases and deaths until April 10, as well as their approximate first report date of Covid-19 (Table 1). From them, it was possible to calculate the $R_1$ and $R_2$ of Covid-19 cases and deaths of 170 countries; it was not possible to estimate neither of the ratios in 40 countries due to limited data available for these countries. The $R_1$ and $R_2$ ranged from 0·20 to 6·64 and from 0·17 to 22·06, respectively. Our calculation showed that 73 countries (42·94%) are in the epidemic phase A (initial accelerated phase), 57 (33·52%) were decelerating getting closer to the peak (phase B), and 40 (23·5%) were already on the other side of the epidemic curve (phase D and E). For deaths, the $R_1$ was calculated for 116 countries with a range between 0 and 6, while for the $R_2$ only in 96 (range, 0 to 32·7). The majority of them, 74 (66·07%), showed an increase in mortality ratios ($R_1$ and $R_2$). Due to absent data on mortality, 75 and 112 countries lacked $R_1$ and $R_2$ calculation, respectively. As an overall, the world is in A and B epidemic phases and is increasing death rates (45·8%) (Table 1).

From the 20 countries with highest $R_1$ (range, 2·38 to 6·64) and $R_2$ (range 2·20 to 22·06) for Covid-19 new cases the majority were from Africa (around 40%, Fig 2) and had a middle-income economy (70% for $R_1$ and 35% for $R_2$). In case of deaths the range of $R_2$ were from 2·73 to 6 and for $R_2$ from 1·33 to 32·70. The predominant region for both ratios was Europe, while high income countries were predominant for the $R_1$ and middle-income for $R_2$ (Table 2). In general terms, from all the countries with high $R_1$ low income countries represented 9·92% (13 countries) and 3·75% (3 countries) for cases and deaths respectively, whereas for high $R_2$ 12·79% (11 countries) for cases and 5·71% (2 countries) for deaths.

We then calculated the median $R_1$ and $R_2$ rates for new cases according to the epidemic phase (Fig 1). While $R_1$ median rate starts with 1·69 (for phase A) and then decreases to 0·79 in phase E; $R_2$ median rate starts with 1·42 for phase A and decreases to 0·81 in phase B but then increases again in phase D to 1·23 to then decrease in phase E to 0·75 thus being aligned with Farr's law.

**Correlational analyses.** Regarding the correlation analysis, the $R_1$ for mortality was positively correlated with urban index (rho = 0·2, p = 0·03), and with deaths per 100 000 inhabitants (rho = 0·3, p = 0·001). No significant correlations were found for the rest of the ratios.

**Predictive analyses.** For the prediction of new cases, we included 69 countries out of 210 countries, the rest of them did not meet the assumption criteria or have enough data. On the

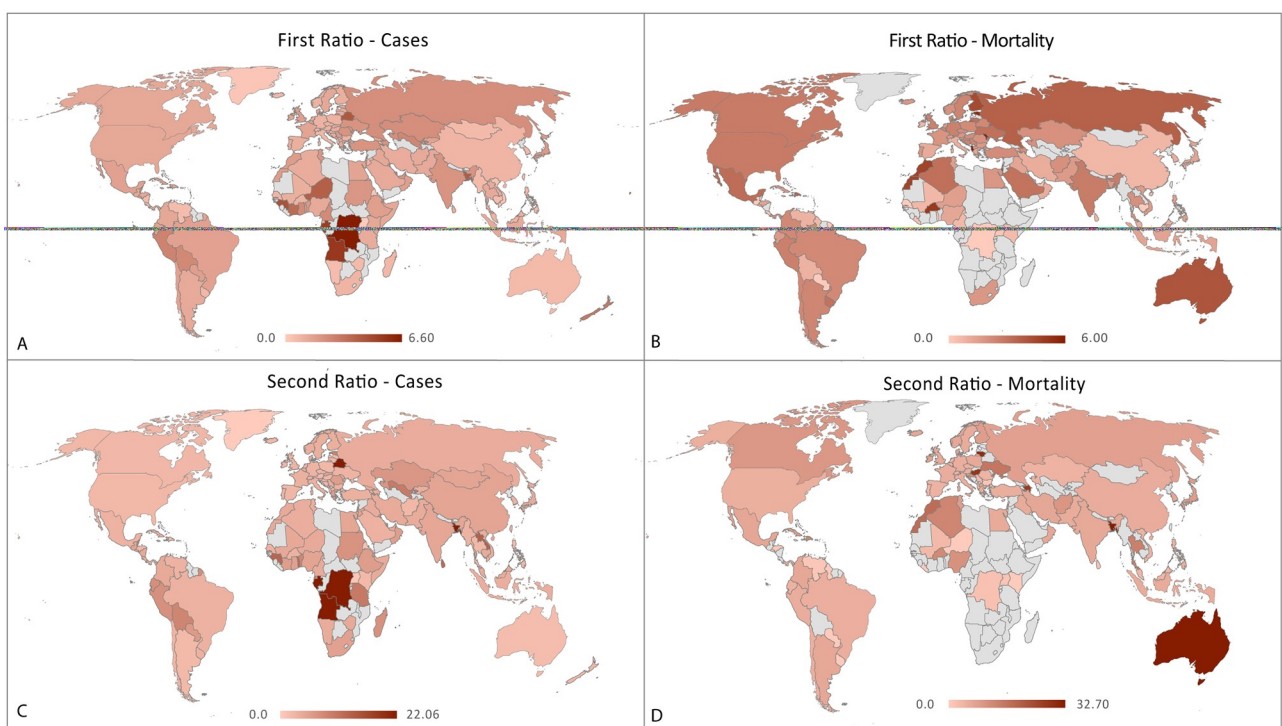

**Fig 2. Geographical representation of Farr's ratios of the COVID-19 pandemic.** First ratios represent a percent increase in the epidemic dynamic, and second ratios represent epidemic acceleration. A and B depict the first ratios for the new cases and deaths, respectively. C and D represent the second ratios for new cases and deaths, respectively. The calculations are based on worldwide data until April 10, 2020.

other hand, for the prediction of mortality, 64 countries were included in the modeling (in S2 Table and in S3 Table).

Worldwide we predict 1 284 553·6 (CI 95%, 935 337·5–1 988 290·9) of new cases (43·1% of the total cases) and 221 329·3 (CI 95%, 155 105·3–371 461·1) new deaths (68·1% of the total deaths) during the period after April 10 to June 10. The peak of new cases would reach around April 11 to 15th with approximately 432 4843·7 new cases (CI 95%, 400 294·6–464 672·7) and the peak of mortality around April 16 to 20th with approximately 46 051·7 deaths during this period (CI 95%, 39 846·2–52 870·2). Following a bell-shape curve, regardless neither of the new cases and new deaths reach zero until June 10, the lowest number of new cases would be around 1·2 (CI 95%, 0–321·3) new cases and 38 (CI 95%, 0·9–1 378·8) new deaths during the lowest peak on June 6th -10th. (Fig 3, S2 and S3 Tables).

Regarding the prediction of individual countries, we divided them into quartiles based on the number of daily cases and deaths trends (Figs 4 and 5). The highest quartile of new cases has a range of 1 863 to 165 364 daily cases and included 18 countries (56% from Europe, 22% from America, 17% from Asia and 6% from Oceania). The highest quartile for mortality includes values from 141 to 3 2867 daily deaths with 16 countries (10 [62%] from Europe, 4 [25%] from America, 1[6%] from Asia, and 1[6%] from Africa). Regarding new cases, from 69 countries, 56 will reach zero, and 13 will continue beyond June 10, 2020 (see S2 Table). For new deaths, from 64 countries, 58 will reach zero deaths and 6 will still be continued beyond June 10, 2020 (in S3 Table). The countries with higher predicted cases are the USA, UK, and Spain, and higher predicted mortality are the USA, France, and Sweden (Figs 4 and 5 and S2 and S3 Tables).

**Table 2. Countries with higher Farr's ratios associated with COVID-19 pandemic.**

| Order | Country | Income | Population Density (inhab/km²) | Cases Adults ≥65y (%) | Cases Total | Cases First Ratio | Epidemic status on June 10 | New Cases on June 10 | Order | Country | Income | Population Density (inhab/km²) | Deaths Adults ≥65y (%) | Deaths Total | Deaths First Ratio |
|---|---|---|---|---|---|---|---|---|---|---|---|---|---|---|---|
| 1 | Congo | Lower Middle | 15·36 | 2.68 | 60 | 6·64 | Descend not clear | 45^ | 1 | Albania | Upper Middle | 104·61 | 13.74 | 23 | 6·00 |
| 2 | Angola | Lower Middle | 24·71 | 2.22 | 19 | 5·63 | Increasing | 17 | 2 | Moldova | Lower Middle | 123·52 | 11.47 | 29 | 5·85 |
| 3 | Gabon | Upper Middle | 8·22 | 3.56 | 44 | 5·38 | Descend not clear | 47 | 3 | Estonia | High | 30·39 | 19.63 | 24 | 5·50 |
| 4 | Guinea | Low | 50·52 | 2.93 | 194 | 4·61 | Descend not clear | 42^ | 4 | Luxembourg | High | 250·09 | 14.18 | 54 | 5·02 |
| 5 | Belarus | Upper Middle | 46·73 | 14.85 | 1,981 | 4·35 | Plateau | 801 | 5 | Burkina Faso | Low | 72·19 | 2.41 | 24 | 5·00 |
| 6 | Bangladesh | Lower Middle | 1239·58 | 5.16 | 424 | 4·34 | Increasing | 3190 | 6 | Israel | High | 410·53 | 11.98 | 95 | 4·80 |
| 7 | Niger | Low | 17·72 | 2.6 | 410 | 4·06 | Descending | 0 | 7 | Morocco | Lower Middle | 80·73 | 7.01 | 107 | 4·61 |
| 8 | Sint Maarten | High | 1235·29 | - | 50 | 3·82 | No cases* | 0 | 8 | Finland | High | 18·16 | 21.72 | 48 | 4·37 |
| 9 | Gibraltar | High | 3371·80 | - | 127 | 3·81 | Descending | 0 | 9 | Australia | High | 3·25 | 15.66 | 53 | 4·07 |
| 10 | Ivory Coast | Lower Middle | 78·83 | 2.86 | 444 | 3·74 | Increasing | 186 | 10 | San Marino | High | 563·08 | - | 34 | 3·64 |
| 11 | New Zealand | High | 18·55 | 15.65 | 1,283 | 3·16 | No cases* | 0 | 11 | Russia | Upper Middle | 8·82 | 14.67 | 94 | 3·61 |
| 12 | Fiji | Upper Middle | 48·36 | 5.45 | 16 | 3·00 | No cases* | 0 | 12 | Dominican Republic | Upper Middle | 219·98 | 7.08 | 126 | 3·50 |
| 13 | Qatar | High | 239·59 | 1.37 | 2,512 | 2·76 | Descend not clear | 1716 | 13 | Mexico | Upper Middle | 64·91 | 7.22 | 194 | 3·20 |
| 14 | Peru | Upper Middle | 24·99 | 8.09 | 5,897 | 2·76 | Descend not clear | 5087 | 14 | Lithuania | High | 44·53 | 19.71 | 22 | 3·11 |
| 15 | Uzbekistan | Lower Middle | 77·47 | 4.42 | 624 | 2·73 | Increasing (2nd peak) | 103 | 15 | Algeria | Upper Middle | 17·73 | 6.36 | 256 | 3·10 |
| 16 | Bolivia | Lower Middle | 10·48 | 7.19 | 268 | 2·56 | Increasing | 695 | 16 | Ukraine | Lower Middle | 77·03 | 16.43 | 69 | 3·09 |
| 17 | Ghana | Lower Middle | 130·82 | 3.07 | 378 | 2·52 | Descend not clear | 291^ | 17 | Andorra | High | 163·84 | - | 26 | 3·06 |
| 18 | Cameroon | Lower Middle | 53·34 | 2.73 | 803 | 2·49 | Descend not clear | 369 | 18 | Saudi Arabia | High | 15·68 | 3.31 | 47 | 3·01 |
| 19 | Russia | Upper Middle | 8·82 | 14.67 | 11,917 | 2·41 | Plateau | 8404 | 19 | Uruguay | High | 19·71 | 14.81 | 7 | 3·00 |
| 20 | Belize | Upper Middle | 16·79 | 4.74 | 10 | 2·38 | Descending | 0 | 20 | USA | High | 35·77 | 15.81 | 18,015 | 2·73 |

(*Continued*)

**Table 2.** (Continued)

| Order | Country | Income | Population Density (inhab/km²) | | Total | Second Ratio | Epidemic status on June 10 | New Cases on June 10 | Order | Country | Income | Population Density (inhab/km²) | | Total | Second Ratio |
|---|---|---|---|---|---|---|---|---|---|---|---|---|---|---|---|
| 1 | Angola | Lower Middle | 24.71 | 2.22 | 19 | 22.06 | Increasing | 17 | 1 | Luxembourg | High | 250.09 | 14.18 | 54 | 32.70 |
| 2 | Bangladesh | Lower Middle | 1239.58 | 5.16 | 424 | 13.66 | Increasing | 3190 | 2 | San Marino | High | 563.08 | - | 34 | 23.48 |
| 3 | Congo | Lower Middle | 15.36 | 2.68 | 60 | 12.19 | Descend not clear | 45^ | 3 | Australia | High | 3.25 | 15.66 | 53 | 12.59 |
| 4 | Dominica | Low | 95.50 | - | 16 | 10.25 | Descending | 0 | 4 | Bangladesh | Lower Middle | 1239.58 | 5.16 | 27 | 8.50 |
| 5 | Gabon | Upper Middle | 8.22 | 3.56 | 44 | 9.62 | Descend not clear | 47 | 5 | Lithuania | High | 44.53 | 19.71 | 22 | 4.53 |
| 6 | Belarus | Upper Middle | 46.73 | 14.85 | 1,981 | 9.17 | Plateau | 801 | 6 | Hungary | High | 107.91 | 19.16 | 77 | 4.28 |
| 7 | Liechtenstein | High | 236.94 | - | 79 | 6.54 | No cases* | 0 | 7 | Azerbaijan | Upper Middle | 120.27 | 6.20 | 10 | 4.13 |
| 8 | Gibraltar | High | 3371.80 | - | 127 | 6.48 | Descending | 0 | 8 | Morocco | Lower Middle | 80.73 | 7.01 | 107 | 2.67 |
| 9 | Saint Lucia | Low | 298.18 | 9.81 | 14 | 5.00 | Descending | 0 | 9 | Ukraine | Lower Middle | 77.03 | 16.43 | 69 | 2.41 |
| 10 | Fiji | Upper Middle | 48.36 | 5.45 | 16 | 4.00 | No cases* | 0 | 10 | Singapore | High | 7953.00 | 11.46 | 7 | 2.13 |
| 11 | Guadeloupe | . | 245.70 | - | 143 | 3.46 | Descending | 0 | 11 | Thailand | Upper Middle | 135.90 | 11.90 | 33 | 2.11 |
| 12 | Laos | Lower Middle | 30.60 | 4.08 | 16 | 3.13 | No cases* | 0 | 12 | Mauritius | Upper Middle | 623.30 | 3.14 | 9 | 2.00 |
| 13 | Guinea | Low | 50.52 | 2.93 | 194 | 2.93 | Descending | 42^ | 13 | Algeria | Upper Middle | 17.73 | 6.36 | 256 | 2.00 |
| 14 | Togo | Low | 145.05 | 2.87 | 76 | 2.89 | Descend not clear | 21 | 14 | Dominican Republic | Upper Middle | 219.98 | 7.08 | 126 | 1.93 |
| 15 | Sri Lanka | Upper Middle | 345.56 | 10.47 | 190 | 2.84 | Descending | 10 | 15 | Burkina Faso | Low | 72.19 | 2.41 | 24 | 1.84 |
| 16 | Tanzania | Low | 63.58 | 2.6 | 32 | 2.42 | No cases* | 0 | 16 | Armenia | Upper Middle | 103.68 | 11.25 | 12 | 1.69 |
| 17 | Uzbekistan | Lower Middle | 77.47 | 4.42 | 624 | 2.35 | Increasing (2nd peak) | 103 | 17 | Nigeria | Lower Middle | 215.06 | 2.75 | 7 | 1.61 |
| 18 | Equatorial Guinea | Upper Middle | 46.67 | 2.46 | 18 | 2.22 | No cases* | 0 | 18 | Afghanistan | Low | 56.94 | 2.58 | 15 | 1.50 |
| 19 | Sint Maarten | High | 1235.29 | - | 50 | 2.21 | No cases* | 0 | 19 | Croatia | Upper Middle | 73.08 | 20.45 | 21 | 1.49 |
| 20 | Estonia | High | 30.39 | 19.63 | 1,258 | 2.20 | Descending | 11 | 20 | Canada | High | 4.08 | 17.23 | 556 | 1.46 |

inhab: inhabitants; USA: United States of America.

*No cases in more than 10 days.

^Cases from June 09, no cases were reported on June 10.

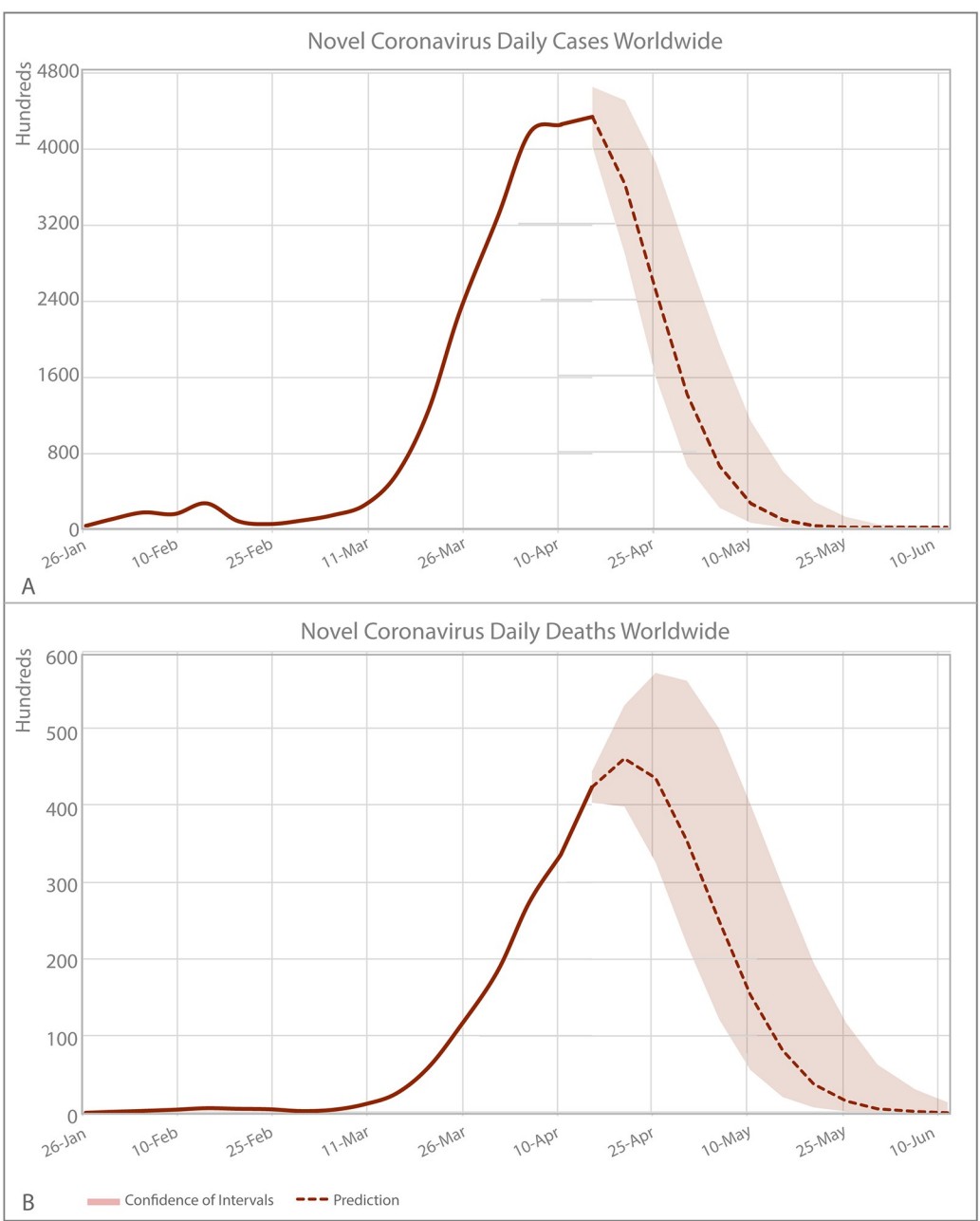

**Fig 3. Worldwide COVID-19 new cases and deaths incidence predicted by Farr's law.** (A) New cases and (B) deaths incidence (and 95% CI). The calculations are based on worldwide data until April 10, 2020.

**Comparison with updated data.** We found in our post-hoc comparison with updated data (June 10, 2020) that from the countries we predicted a higher number of cases and deaths worldwide (using data till April 10, 2020), 70% and 100% are actually among the first 20 countries with more cases and deaths, respectively, by June 2020. Our model predicted high dynamics in US, UK, Brazil, Italy, Spain, France (Table 1), and this was confirmed with current data. Additionally, we found 55 (26.2%) countries reported strict restriction strategies as part of political actions against the pandemic, the most common were stay-home policies, gathering

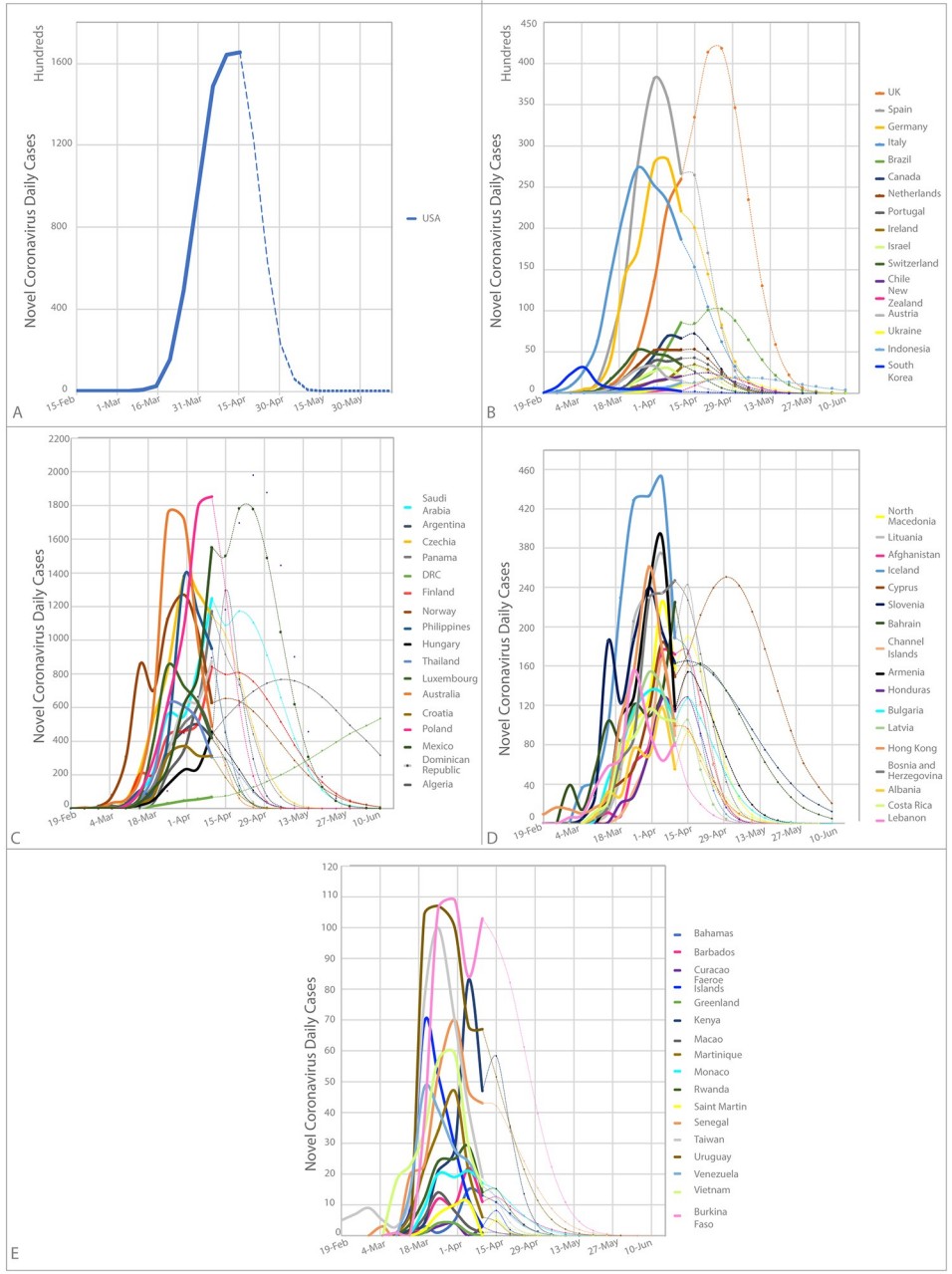

**Fig 4. Prediction of new cases per country predicted by Farr's law.** A) USA prediction—the country with a higher incidence in the world. B) Prediction for quartile 1. C) Prediction for quartile 2. D) Prediction for quartile 3. E) Prediction for quartile 4. The quartile division is based on the number of new cases. Only 70 countries are included in this prediction analyses, due to the lack of data and failure to meet the assumptions criteria. The calculations are based on worldwide data up to April 10, 2020.

restriction, and travel restrictions (S4 Table). However, there is important missing information for several countries in the available dataset (IHME).

Regarding the worldwide curve for new cases and deaths of Covid-19 in June 2020 (S1 Fig) showed a pseudo-normal distribution (negative kurtosis), with a steep slope by the second semester of March with a plateau by the beginning of April which last one month. By the

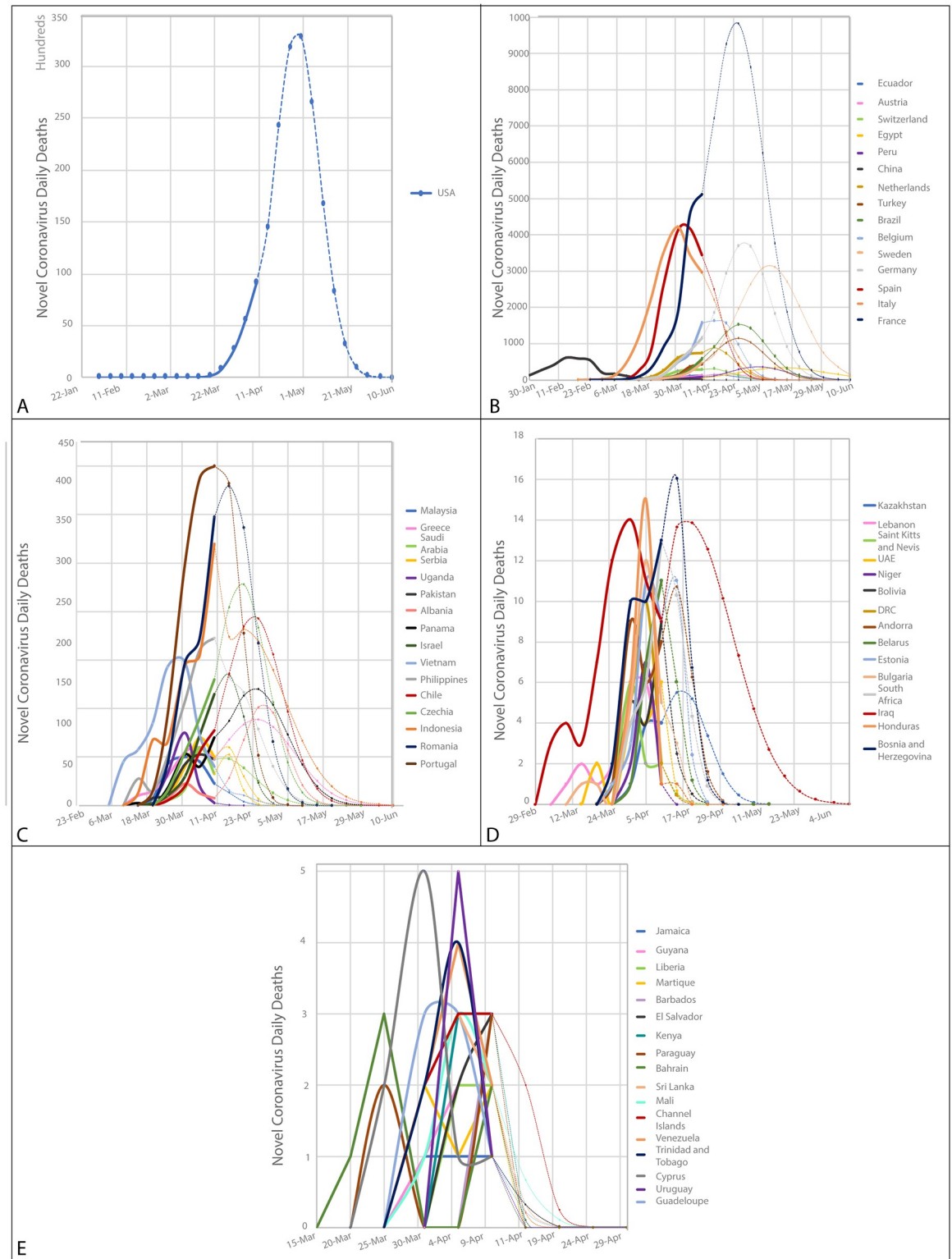

**Fig 5. Prediction of new deaths per country predicted by Farr's law.** A) USA prediction—the country with a higher incidence in the world. B) Prediction for quartile 1. C) Prediction for quartile 2. D) Prediction for quartile 3. E) Prediction for quartile 4. The quartile division are based on the number of deaths. Only 68 countries are included in this prediction analyses, due to the lack of data and failure to meet the assumptions criteria. The calculations are based on worldwide data up to April 10, 2020.

beginning of May the curve started to rise again but gradually. Similarly, the curve of death had a steep slope during the last 10 days, from March reaching a peak by mid-April, and then a gradual descent until the end of May, where it reached a plateau.

By June 10, 36 countries (64.6%) of the 56 countries we predicted to be around no new cases before June 10 are decreasing or around zero new daily cases (Table 1). Moreover, as we predicted countries as New Zealand, Greenland, Macao, Saint Martin, and Faeroe Islands report no new cases for more than ten days up to June 10.

Concerning the 20 countries with high R1 for new Covid-19 cases (Table 2), we found that five countries (25%) were still having an increment on their curve with one of them (Uzbekistan) increasing a second wave. Two countries (10%) were in a plateau; for seven (35%), the descend on the curve was still not clear, while three (15%) has a clear descend and other three (15%) has not reported new cases for more than ten days. Among the three countries with the highest R1, Congo and Gabon are still reporting cases with a heterogeneous pattern, which hinders the determination of a clear dynamic on the curve. On the other hand, Angola reported its peak on June 10 and still in an accelerated phase of the pandemic (Table 2).

In the case of the countries with high R2 for cases, three countries (15%) had an increasing curve, one (5%) was in a plateau, three (15%) had a not clear descend, seven (35%) were descending and six (30%) had not reported cases for more than ten days. Among the countries with the highest R2, Angola and Bangladesh showed a clear increment on its curve, reporting more new cases on June 10 (Table 2).

## Discussion

Farr's law is a simple arithmetical model that provides useful and important insights on epidemic dynamics. The findings from our modeling suggest that most of the countries over the world (76·43%) are in the early stage of the epidemic curve (phase A and B of our theoretical framework). The countries with higher epidemic dynamics (acceleration of cases and death numbers) are in Africa (around 40%) and had middle-income economies. Based on our model, the pandemic curve will reduce significantly until June 10, 2020, for both new cases and deaths, in the overall worldwide model and for 56 countries (in S2 and S3 Tables). The countries with higher predicted cases (adjusted for population) are USA, UK, and Spain, and higher predicted mortality are USA, France and Sweden; however, 60% of the countries could not enter to the predictive modeling due to lack of data or instability of $R_2$ estimate (higher than 1).

The percentage of countries on phases A and B and with higher dynamics from low and middle-income sectors are higher (mainly from Africa and Latin America). This is a potential risk due to the limited health resources in those countries that could lead to a high rate of mortality and burden for the health system but also could generate a devastating socioeconomic, political, and inequality impact [19, 20]. Recent studies are reporting the lack of preparedness and high vulnerability of African countries against an eventual increase of COVID-19 cases [5]. Also, Moore et al. reported a ranking of countries based on the infectious disease vulnerability index [21], which considered a number of socioeconomic and health factors, several countries from the top of their list, such as Angola, Niger, Guinea, Congo, Togo, and Ivory Coast are in our ranking using the Farr's ratios, indicating a higher epidemic dynamics in those countries, yet with a small number of cases, currently. Thus, this is a call to prioritize actions in those countries to intensify surveillance, to re-allocate resources, and to build healthcare capacities based on multi-nation collaboration [22] to limit onward transmission and to reduce the future impact on these regions.

Based on our prediction, the worldwide trend will reach values near to zero at the beginning in June, and approximately 56 countries (S2 Table) will reach values near zero before June 10,

2020. Compared with the current trend at June 10, 2020, we can see a pseudo-normal distribution with low kurtosis (more pronounced in the curve of new cases–S1 Fig). This could be explained by the heterogeneity of the clusters (countries) included in the model, with different pandemic start date, different socioeconomic characteristics, and public health and political actions against the pandemic; therefore, this produces potentially an overlap of multiple normal distributions curves. This also could be true for large countries with independent states such as USA (implementing multiple political actions and public health strategies at different moments) [23]. However, for more homogenous clusters (such as New Zealand, Australia, and some Asian countries), the predicted curves were accurate. Similarly, the prediction estimates were also accurate—most of the countries (70%) that we predicted a higher number of cases and deaths (till June 10) were confirmed in our post-hoc extraction, as well for the predicted countries with high dynamics (higher predicted R1 and R2). Thus, we should consider that our estimates using the Farr's law depend on the precision of the reported data, the cluster heterogeneity, and the current acceleration ($R_2$) of the epidemic dynamics (i.e. higher values of $R_2$ produces an exponential function of the fitted values). Similar behavior was reported in previous studies [11, 12]. Although, Santillana et al. suggested a potential use of these higher $R_2$ ratios, not to use them as predictive measures but rather as sentinel index for the change in epidemic dynamic, which could indicate the start of a new wave of cases [11].

Previous studies have reported behavior predictions for the COVID-19 pandemic; most of them focus on specific countries, such as China [24], Chile [25], Italy [24], France [24], and USA [23, 26]. The estimation of end date varies from May 12 (for Chile) to June 15 (for Italy), these dates from more complex models (most of them from a SIR model) are consistent with our predictions for those countries (Table 1), suggestion an acceptable accuracy to describe the epidemic dynamics with a simpler model. None of the previous studies used the Farr's law to model the current pandemic behavior, and the available models reported prediction for high-income countries with better health system infrastructure and data registration; however, we could not identify published models from low- and middle-income countries, especially from Africa, those who are paradoxically more at risk due to high pandemic dynamic. Thus, Farr's law approximation will be a valid option for scenarios with low resources and to identify countries at risk.

Moreover, it has been reported Farr's law is an adequate model to assess the behavior of epidemics more than predict the exact number of cases accurately [2]. However, under certain assumptions (in epidemic phases with relative deceleration), its estimates are near to the SIR and IDEA models [11]. Besides, it seems to apply across different outbreaks types—because it relies on the intrinsic natural history of epidemics—and allow as to model fast with simple assumptions and limited data. The $R_1$ and $R_2$ ratios are variable across countries and epidemic phases, allowing us to classify the epidemic behavior over the world. Besides, it does seem that a higher $R_1$ for mortality is associated with a high urban index and a higher number of deaths per 100,000 inhabitants, which is along with the literature on the impact of urbanization on the transmission of respiratory infection diseases [11]. Therefore, countries could also use $R_1$ and $R_2$ ratios to monitor the first deceleration phase (Phase B). Interestingly, the median $R_2$ ratio is similar to the past AIDS epidemic reports [9], thus reflecting perhaps the behavior of an outbreak without immune protection.

The sociodemographic characteristics and the political actions against the pandemic are important factors within countries to describe pandemic behavior. From our model, we predicted 43 countries (Table 1) would reach near to zero in June. Most of them are middle to high-income countries, implement early and strict restriction policies; it seems no particular sociodemographic characteristics (population size, density, or proportion of older people) are predominant in these groups. These findings are aligned with previous studies showing the

positive effects of strict restriction policies [27]. From our model and post-doc extraction, countries as New Zealand reached no new cases around May, while Australia has less than 20 new daily cases by June 10. The normality of these curves might be related to different explanations. First of all, both countries are high-income countries with a high Human Capital Index (0.8 and 0.7 respectively) that have invested in health since 1990 [28]. Germany, with 21.5% of the population more than 65 years old, has a smaller number of deaths per million people than other countries like US, Italy, Spain. All these countries established political and health social regulations during the pandemic and widespread testing even before reaching the peak. However, the key factors to successfully implement restrictive measurements could be adequate health literacy and socioeconomic equity [29]. For example, in our model, we found a high dynamic in Peru; it was one of the first countries in America to close its borders, established a strict lock-down with a curfew, and even provided financial aid for vulnerable people during the lock-down [30]. However, high socioeconomic disparities, great urban low-income conglomerations with limited health services accessibility, high levels of business informality [31], and even corruption have played an important role in jeopardizing the fight against the Covid-19 outbreak. Despite that, the Peruvian government's quarantine policies might have prevented a major sanitary catastrophe considering the fragile and fragmented Peruvian public health system [32]. Indeed, our results suggested that restrictive measurements (social-distancing, restricting gatherings and non-essential travels) and widespread testing are critical to ending the ongoing COVID-19 pandemic. However, it is necessary to consider the sociodemographics and health literacy characteristics of the population to implement these measures in the mid- to long-term successfully.

One limitation in our modeling is the probable high rate of underreported cases all over the world, especially in low and middle economies either because of population size, weak health systems, geographical issues, inequity or lack of health access [33]. Additionally, the migration of people between countries is an important covariate for epidemic modeling [34] we did not include in our model due to data availability. In fact, this variable could contribute to underestimating the real dynamic; however, the utility for public health decision still valid as a similar limitation was founded in the previous influenza H1N1 pandemic [35]. Finally, another limitation is the potential heterogeneity on the criteria to identify cases in the Worldometer dataset, since they used a confirmation status based on public health policies and available test in each country. Therefore, there is a possibility of underreporting and false negatives cases [36], especially in low- and middle-income countries.

Finally, even though our model predicts a significant reduction of cases and deaths worldwide, a second wave of cases is possible. Currently, there are examples of countries with these patterns, such as South Korea and China, countries where the restriction strategies and political actions were applied early and rigorously. This highlights the importance of other factors such as viral reintroduction, particularly international importation from countries with higher epidemic dynamics, as well as a rebound of viral transmissibility due to the gradual resumption of economic activities and normal levels of social interaction [37]. In this scenario, our modeling approach could be a potential tool to assess the pandemic dynamics in a simple manner, especially in regions with already compromised health and socioeconomic systems.

In conclusion, to develop a global health perspective on a pandemic, the first step could be to use of simple modeling techniques to depict a broad global picture of the disease's dynamics, that allow us to properly identify areas with high-risk due to high dynamic of the disease. Farr's law seems to be a useful model to give an overview of COVID-19 pandemic dynamics. The regions with high dynamics are countries from Africa and Latin America. Thus, this is a call to urgently prioritize actions in those countries to intensify surveillance and re-allocate resources based on multi-nation collaboration to limit onward transmission and to reduce the

future impact on these regions. Close monitoring of epidemic dynamics is needed to ensure correct worldwide policy interventions and to be prepared for an eventual COVID-19 second wave.

## Supporting information

**S1 Table. GATHER checklist.**
(DOC)

**S2 Table. Cases predicted.**
(XLSX)

**S3 Table. Deaths predicted.**
(XLSX)

**S4 Table. Restriction policies.**
(DOCX)

**S1 Fig. Worldwide COVID-19 new cases and deaths until June 10.** (A) New daily cases and (B) new daily deaths incidence.
(TIF)

## Author Contributions

**Conceptualization:** Kevin Pacheco-Barrios, Alejandra Cardenas-Rojas, Stefano Giannoni-Luza, Felipe Fregni.

**Formal analysis:** Kevin Pacheco-Barrios, Felipe Fregni.

**Investigation:** Kevin Pacheco-Barrios, Alejandra Cardenas-Rojas, Stefano Giannoni-Luza, Felipe Fregni.

**Methodology:** Kevin Pacheco-Barrios, Alejandra Cardenas-Rojas, Stefano Giannoni-Luza, Felipe Fregni.

**Validation:** Kevin Pacheco-Barrios.

**Visualization:** Alejandra Cardenas-Rojas, Stefano Giannoni-Luza.

**Writing – original draft:** Kevin Pacheco-Barrios, Alejandra Cardenas-Rojas, Stefano Giannoni-Luza, Felipe Fregni.

**Writing – review & editing:** Kevin Pacheco-Barrios, Alejandra Cardenas-Rojas, Stefano Giannoni-Luza, Felipe Fregni.

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
