## [Decision Letter · Decision Letter 0]

5 Jun 2020

PONE-D-20-12267

COVID-19 Pandemic and Farr's Law: a global comparison and prediction of outbreak acceleration and deceleration rates

PLOS ONE

Dear Dr. Fregni,

Thank you for submitting your manuscript to PLOS ONE. After careful consideration, we feel that it has merit but does not fully meet PLOS ONE’s publication criteria as it currently stands. Therefore, we invite you to submit a revised version of the manuscript that addresses the points raised during the review process.

ACADEMIC EDITOR: This manuscripts is important and subject is relevant .Please make sure to address all comments made by reviewers, specifically the comment mentioned on the source and quality of data. I would also like to see the explicit answer to the question number 6 from the second reviewer . 

We look forward to receiving your revised manuscript.

Kind regards,

Amir Radfar, MD,MPH,MSc,DHSc

Academic Editor

PLOS ONE

Journal Requirements:

3. Please ensure that the manuscript's formatting and style are in line with PLOS ONE guidelines, please see https://journals.plos.org/plosone/s/submission-guidelines for more information. Specifically, please ensure that the information included in the introduction is relevant to the context of the study.

4. We note that Figure 2 in your submission contain map images which may be copyrighted. All PLOS content is published under the Creative Commons Attribution License (CC BY 4.0), which means that the manuscript, images, and Supporting Information files will be freely available online, and any third party is permitted to access, download, copy, distribute, and use these materials in any way, even commercially, with proper attribution. For these reasons, we cannot publish previously copyrighted maps or satellite images created using proprietary data, such as Google software (Google Maps, Street View, and Earth). For more information, see our copyright guidelines: http://journals.plos.org/plosone/s/licenses-and-copyright.

a) You may seek permission from the original copyright holder of Figure 2 to publish the content specifically under the CC BY 4.0 license. 

Reviewers' comments:

Reviewer's Responses to Questions

**Comments to the Author**

1. Is the manuscript technically sound, and do the data support the conclusions?

Reviewer #1: Yes

Reviewer #2: Partly

Reviewer #3: Yes

2. Has the statistical analysis been performed appropriately and rigorously? 

Reviewer #1: Yes

Reviewer #2: Yes

Reviewer #3: Yes

3. Have the authors made all data underlying the findings in their manuscript fully available?

Reviewer #1: Yes

Reviewer #2: Yes

Reviewer #3: Yes

4. Is the manuscript presented in an intelligible fashion and written in standard English?

Reviewer #1: Yes

Reviewer #2: Yes

Reviewer #3: Yes

5. Review Comments to the Author

Reviewer #1: Abstract:

Page 8- Need expansion of COVID-19

Need a brief explanation of why Farr’s laws used in the declaration of rates?

Since the generalization is not possible, region can be mentioned in the title of the paper

Introduction:

Please elaborate on the limitations of the Farr's law and its use in models

Materials and methods:

Isn’t there a need to consider the mobility of the population which has a high potential to alter the changes of cases especially in the case of infectious diseases such as COVID-19

On what criteria used to identify cases, like socio-demographic information.

Under the data author need not keep the URL link of Worldometer since, It is already provided in the reference l

Results:

62% from Europe, 25% from America, 6% from Asia and 6% from America-Here America is repeated twice (USA v/s rest of america, north or south, ? need to explain)

If actual numbers mentioned with percentage, it would give better understanding.

The cases of countries which mostly recovered from COVID can be mentioned and a comparison of the actual time of recovery and that predicted by this model can be mentioned. This can also be mentioned in the tables across to the predicted date as well as the actual date of the last case reported in that particular country.

Discussion:

Findings from other models, differences in the obtained findings compared to other models could be elaborated in discussion, pointing out the added value of using Farr’s law.

Comparison of countries depending on age demographics, recovery time can be included in discussion.

Reviewer #2: Reviewer: Dr. Farshad Farzadfar

Fregni et al. provided a study to investigate Farr’s law in the COVID-19 pandemic. The authors tried to come up with an efficient model to predict acceleration and deceleration rates of the COVID-19 outbreak and probable time of outbreak relative resolution.

Although the authors inspected mentioned notion well, several issues are necessary to be considered, including:

1. In the abstract, it is mentioned that COVID-19 data until April 10, 2020, is utilized for modeling, wasn’t is possible to use newer data for this study?

2. In the introduction, the authors mention that the importance of this study is to prepare the medical system against epidemics. Still, the final results of the study may implicate deceleration and endpoints of the epidemic more prominent. The main goal of the study should be highlighted.

3. Telling the life story of Mr. Farr in the introduction seems to be unnecessary for understanding the aim of this study. Besides, introducing components of Farr’s law in the introduction instead of the methods part can be more useful.

4. The first and second ratios (R1 and R2) are explained vaguely in the methods.

5. Are Worldometer website data on COVID-19 statistics reliable enough to use in such a study? Is there any previous evidence one quality of their statistics? Data quality? Different criteria for diagnosis?

6. Does the “predefined hypothesis that higher R1 and R2 ratios are correlated with higher numbers of cases or deaths” sentence has a reference? Especially about predicting deaths this claim is more questionable.

7. Data presented in S2 and S3 tablets are not consistent with prediction numbers for new cases and new deaths beyond 10th June 2020.

8. The results of the study are not well discussed in the discussion part. More detailed benefits of the results of the study could help readers more, and offer more solutions for health policymakers.

9. Numbers of countries reaching values near zero in new cases and deaths are not discussed that make how much population and talking socioeconomic status of these countries could help more.

10. The conclusion part does not conclude the main message of the study. A warning message for high-risk areas, according to predictions discussed, could be more beneficial.

Reviewer #3: This is an interesting study based on the very old and forgot Farr's law.

I only recommend to comment in the Discussion section the influence of the different political actions against the SARS-CoV-2, and to compare the very symmetric curve of figure 1 with the current epidemiological situation (perhaps in addendum at the lat moment).

6. PLOS authors have the option to publish the peer review history of their article (what does this mean?). If published, this will include your full peer review and any attached files.

Reviewer #1: Yes: Giridhara R Babu, NOLITA DOLCY SALDANHA

Reviewer #2: No

Reviewer #3: No

---

## [Author Response · Author response to Decision Letter 0]

26 Jun 2020

Response Letter

Journal Requirements:

R. Thank you for your comment. We have corrected the style requirements in the revised submission.

R. Thank you for your comment. We have copyedited the revised manuscript. 

3. Please ensure that the manuscript's formatting and style are in line with PLOS ONE guidelines, please see https://journals.plos.org/plosone/s/submission-guidelines for more information. Specifically, please ensure that the information included in the introduction is relevant to the context of the study.

R. Thank you for the suggestion. We have removed the Farr’s biography section to be in line with PLOS ONE guidelines.

4. We note that Figure 2 in your submission contain map images which may be copyrighted. All PLOS content is published under the Creative Commons Attribution License (CC BY 4.0), which means that the manuscript, images, and Supporting Information files will be freely available online, and any third party is permitted to access, download, copy, distribute, and use these materials in any way, even commercially, with proper attribution. For these reasons, we cannot publish previously copyrighted maps or satellite images created using proprietary data, such as Google software (Google Maps, Street View, and Earth). For more information, see our copyright guidelines: http://journals.plos.org/plosone/s/licenses-and-copyright.

R. Thank you for your feedback. We have modified the figure 2 using the natural earth map (public domain, http://www.naturalearthdata.com/) modified and edited with Illustrator by the authors. 

Reviewers' comments:

Reviewer #1: 

Abstract:

Need expansion of COVID-19. Need a brief explanation of why Farr’s laws used in the declaration of rates? 

R. Thank you for your suggestions. We have added in the abstract an expansion on Farr’s law was used, as you suggested. However, due to word limit (until 300 words), we could not detail on COVID-19, although, we consider just a brief introduction is adequate as this is a trending topic worldwide.

Since the generalization is not possible, region can be mentioned in the title of the paper 

R. Thank you for your comment. We agreed that is not possible the generalization of the calculated acceleration and deceleration rates from one region to other, for that we reported the results by country in the table 1. However, since we included all the data available from 210 countries at that moment, we think it is appropriate to mention in the title: a global comparison. 

Introduction: 

Please elaborate on the limitations of the Farr’s law and its use in models 

R. Thank you for your suggestion. We have added in introduction and discussion a section on the limitations of Farr’s law for epidemic modelling. 

Materials and methods: 

Isn’t there a need to consider the mobility of the population which has a high potential to alter the changes of cases especially in the case of infectious diseases such as COVID-19.

R. Thank you for your feedback. We definitely agree with your comment, the internal and external migration patterns can affect the epidemic modelling, it is a recognized confounder for most of available epidemic models (1, 2), also the worldwide data on migration is not available hindering the inclusion of this variable in the models. We have added this factor as limitation in the discussion of this paper. 

1. Greenaway C, Gushulak BD. Pandemics, migration, and global health security. Handbook on migration and security: Edward Elgar Publishing; 2017.

2. Chakraborty I, Maity P. COVID-19 outbreak: Migration, effects on society, global environment, and prevention. Science of the Total Environment. 2020:138882. 

On what criteria used to identify cases, like socio-demographic information. 

R. Thank you for your comment. The socio-demographic information was based on the world bank data repository (1), as we mentioned in the Methods section (Data subsection). The criteria to identify cases was based on Worldometer report, they used a confirmation status based on public health policies of each country, therefore, there is a possibility of underreport and false negatives especially in low- and middle-income countries, as we recognized in our discussion section (as limitation).

1. World Bank. 2019 [cited 2020]. Available from: https://data.worldbank.org/.

Under the data author need not keep the URL link of Worldometer since, it is already provided in the reference l 

R. Thank you for your suggestion. We have deleted the URL from the method section.

Results:

62% from Europe, 25% from America, 6% from Asia and 6% from America-Here America is repeated twice (USA v/s rest of America, north or south, ? need to explain). If actual numbers mentioned with percentage, it would give better understanding. 

R. Thank you for your comment. We have corrected this sentence (it was a typo) and added the absolute number together with the percentages: “10 (62%) from Europe, 4(25%) from America, 1(6%) from Asia, and 1(6%) from Africa” 

The cases of countries which mostly recovered from COVID can be mentioned and a comparison of the actual time of recovery and that predicted by this model can be mentioned. This can also be mentioned in the tables across to the predicted date as well as the actual date of the last case reported in that particular country. 

R. Thank you for your suggestions. We have added in results and table 1 the list of predicted countries mostly recovered from COVID, and a comparison with the current number of cases for those countries, as you recommended. 

Discussion:

Findings from other models, differences in the obtained findings compared to other models could be elaborated in discussion, pointing out the added value of using Farr’s law. Comparison of countries depending on age demographics, recovery time can be included in discussion. 

R. Thank you for your feedback. As you suggested, we have added a comparison with other COVID-19 prediction models in the discussion of this paper. Moreover, we discussion the age demographics and recovery time of the predicted countries mostly recovered from COVID in our model. 

Reviewer #2: 

Fregni et al. provided a study to investigate Farr’s law in the COVID-19 pandemic. The authors tried to come up with an efficient model to predict acceleration and deceleration rates of the COVID-19 outbreak and probable time of outbreak relative resolution. Although the authors inspected mentioned notion well, several issues are necessary to be considered, including:

1. In the abstract, it is mentioned that COVID-19 data until April 10, 2020, is utilized for modeling, wasn’t is possible to use newer data for this study?

R. Thank you for your comment. Since the objective of this study is the exploration of the COVID-19 pandemic dynamic (acceleration and deceleration) using Farr’s law approximation and to identify areas with high dynamic instead to calculate a precise predictions, we consider that the data until April 10, 2020 is adequate for the study goal. Additionally, due to the rapid changes of cases number each day, in our opinion a modelling with updated information is not feasible, however, we have decided to add a in results and table 1 the current number of cases for those countries with predicted values near to zero, and a comparison of the curve in figure 1 with the current worldwide epidemic curve at 06/16/20 (we have added this updated curve as supplementary figure 3).

2. In the introduction, the authors mention that the importance of this study is to prepare the medical system against epidemics. Still, the final results of the study may implicate deceleration and endpoints of the epidemic more prominent. The main goal of the study should be highlighted. 

R. Thank you for your feedback. We have highlighted the main goal of the study in the introduction and discussion (to describe the epidemics dynamics and make predictions to help further preparation of health system in areas with high dynamic). 

3. Telling the life story of Mr. Farr in the introduction seems to be unnecessary for understanding the aim of this study. Besides, introducing components of Farr’s law in the introduction instead of the methods part can be more useful. 

R. Thank you for your comment. We have removed the Farr’s biography section and added a description of the Farr’s law in the introduction. 

4. The first and second ratios (R1 and R2) are explained vaguely in the methods. 

R. Thank you for your suggestion. We have expanded the explanation of the first and seconds ratios in the method section (calculation of ratios subsection).

5. Are Worldometer website data on COVID-19 statistics reliable enough to use in such a study? Is there any previous evidence one quality of their statistics? Data quality? Different criteria for diagnosis?

R. Thank you for your feedback. Worldometer is composed by a team of researchers, developers, and volunteers with no political, governmental, or corporate affiliation with the aim to provide time relevant world statistics. As general information it has been voted as one of the best free reference websites by the American Library Association and its data has been used in the United Nations Conference Rio+20 (1). 

In the context of COVID-19 pandemic it has been a provider for important governmental institutions such as the UK, Thailand, Pakistan, Sri Lanka, and Vietnam Governments as well as for the John Hopkins CSSE (2). Based on its webpage, they obtain the data directly from official reports from Government´s communication channels or indirectly, through local media sources when they are considered reliable. Besides, they claim to have more than 6000 citations in journal articles form which more than 11 had citied Worldometer data in the context of Covid-19 pandemic. We have identified that at least four articles about the Covid-19 pandemic are published in Q1 journals (including a correspondence letter in Lancet [3]) and five on Q2 journals. We chose to use this data, not only because of its reliability based on the above, but also considering its availability since the objective of this study is the exploration of the COVID-19 pandemic dynamic (acceleration and deceleration) using Farr’s law approximation and to identify areas with high dynamic instead to calculate a precise predictions. Finally, we added these details in method section to clarify the validity of the data source. 

1. Otto-Zimmermann K. From Rio to Rio+ 20: the changing role of local governments in the context of current global governance. Local Environment. 2012;17(5):511-6.

2. Covid C. Dashboard by the Center for Systems Science and Engineering (CSSE) at Johns Hopkins University (JHU). 2020.

3. Flahault A. Has China faced only a herald wave of SARS-CoV-2? The Lancet. 2020;395(10228):947.

6. Does the “predefined hypothesis that higher R1 and R2 ratios are correlated with higher numbers of cases or deaths” sentence has a reference? Especially about predicting deaths this claim is more questionable.

R. Thank you for your comment. The initial Farr’s observation and future studies based on his calculations (1, 2) was that the R1 and R2 ratios are correlated with the number of cases, therefore the arithmetical calculation could allow us to infer the future cases dynamic. However, we recognized that the number of deaths are influenced by multiple other factors than the number of cases, which are out of the scope of our analysis. Therefore, we have decided to modify that sentence: “predefined exploratory hypothesis that higher R1 and R2 ratios are correlated with higher numbers of cases”, and to add the corresponded references.

1. Bregnman DJ LA. Farr´s law applied to AIDS projections. JAMA. 1990;263(11):1522-5.

2. Santillana M TA, Nasserie T, Fine P, Champredon D, Chindelevitch L, Dushoff J, Fisman D. Relatedness of the incidence decay with exponential adjustment (IDEA) model, "Farr's law" and SIR compartmental difference equation models. Infect Dis Model. 2018;3:1-12. doi: 10.1016/j.idm.2018.03.001.

7. Data presented in S2 and S3 tablets are not consistent with prediction numbers for new cases and new deaths beyond 10th June 2020. 

R. Thank you for your comment. We have checked the tables 1, S2, and S3. The table S2 and S3 describes the predicted number of cases and deaths, respectively, after April 10th, 2020 until June 10th, 2020. For the 13 countries that we predicted they will have >0 new cases or deaths until the end of our simulation, we categorized them as countries having cases and deaths beyond June 10th.We agree with you that the term “beyond June 10th, 2020” in the table 1 could be confusing, therefore, we added a legend in table 1 explaining that those countries have more than 0 predicted cases or deaths after we end our simulation (June 10th), meaning the will continue with the pandemic wave beyond the end our simulation. 

8. The results of the study are not well discussed in the discussion part. More detailed benefits of the results of the study could help readers more and offer more solutions for health policymakers. (Kevin) 

R. Thank you for your feedback. We have added in the discussion section, a comment on demographic and socioeconomic status of the countries with predicted values near to zero, as well political actions against the SARS-CoV-2 in the countries with higher acceleration rates and those predicted countries mostly recovered from COVID (number of cases near to zero). As you mentioned, this information will be more helpful for health policymakers.

9. Numbers of countries reaching values near zero in new cases and deaths are not discussed that make how much population and talking socioeconomic status of these countries could help more. 

R. Thank you for your suggestion. We have added a discussion paragraph on demographic and socioeconomic status of the countries with predicted values near to zero. 

10. The conclusion part does not conclude the main message of the study. A warning message for high-risk areas, according to predictions discussed, could be more beneficial. 

R. Thank for your feedback. We agree with your suggestion, we have highlighted in the conclusion part the warning message for high-risk areas. 

Reviewer #3: 

This is an interesting study based on the very old and forgot Farr's law.

I only recommend to comment in the Discussion section the influence of the different political actions against the SARS-CoV-2, and to compare the very symmetric curve of figure 1 with the current epidemiological situation (perhaps in addendum at the lat moment).

R. Thank you for your feedback. As you suggested, we have added in our discussion section the different political actions against the SARS-CoV-2 in the countries with higher acceleration rates and those predicted countries mostly recovered from COVID (number of cases near to zero). Additionally, we have compared the curve of figure 1 with the current worldwide epidemic curve at 06/16/20 (we have added this updated curve as supplementary figure 3).

---

## [Decision Letter · Decision Letter 1]

2 Sep 2020

COVID-19 Pandemic and Farr's Law: a global comparison and prediction of outbreak acceleration and deceleration rates

PONE-D-20-12267R1

Dear Dr. Fregni,

We’re pleased to inform you that your manuscript has been judged scientifically suitable for publication and will be formally accepted for publication once it meets all outstanding technical requirements.

Kind regards,

Amir Radfar, MD,MPH,MSc,DHSc

Academic Editor

PLOS ONE

Additional Editor Comments (optional):

Reviewers' comments:

Reviewer's Responses to Questions

**Comments to the Author**

1. If the authors have adequately addressed your comments raised in a previous round of review and you feel that this manuscript is now acceptable for publication, you may indicate that here to bypass the “Comments to the Author” section, enter your conflict of interest statement in the “Confidential to Editor” section, and submit your "Accept" recommendation.

Reviewer #2: All comments have been addressed

Reviewer #4: All comments have been addressed

2. Is the manuscript technically sound, and do the data support the conclusions?

Reviewer #2: Yes

Reviewer #4: Yes

3. Has the statistical analysis been performed appropriately and rigorously? 

Reviewer #2: Yes

Reviewer #4: Yes

4. Have the authors made all data underlying the findings in their manuscript fully available?

Reviewer #2: Yes

Reviewer #4: Yes

5. Is the manuscript presented in an intelligible fashion and written in standard English?

Reviewer #2: Yes

Reviewer #4: Yes

6. Review Comments to the Author

Reviewer #2: I appreciate the efforts of the authors for making the manuscript more suitable for publication and considering the suggested comments for revision. Here I want to discuss the answers and propose further comments:

C1. I thank the authors for updating the data and doing a part of modeling on the updated data. However, I could not find the mentioned supplementary figure 3 in the attached files and links.

C2. So, the main goal of the study is explained well. You can provide this answer in the introduction part to make the goal of the study clearer.

C3. Thank you for removing the Farr’s biography and describing the law in the introduction.

C4. Expanded explanation of the calculation of R1 and R2 ratios is a positive change, and I thank the authors.

C5. Enough evidence and suitable references are provided for the validity of the Worldometer website and data.

C6. The added reference for the comment and the corrected claim of authors seem to be more logical now in the context of Farr’s law.

C7. The authors did well with adding an explanation to table 1 as legend and make the data and results easier to understand for readers of the article.

C8. Adding the suggested parts to the discussion (as the discussion on demographic and socioeconomic status and political actions of the countries) to help health policymakers is another positive change in the manuscript made by the authors and it is respectful.

C9. The revised discussion part has covered this comment.

C10. A revised conclusion of the study seemed to be necessary to warn the high-risk areas, and I thank the authors for considering this comment in their manuscript.

Reviewer #4: (No Response)

7. PLOS authors have the option to publish the peer review history of their article (what does this mean?). If published, this will include your full peer review and any attached files.

Reviewer #2: No

Reviewer #4: No

---

## [Editor Report · Acceptance letter]

8 Sep 2020

PONE-D-20-12267R1 

COVID-19 Pandemic and Farr's Law: a global comparison and prediction of outbreak acceleration and deceleration rates 

Dear Dr. Fregni:

I'm pleased to inform you that your manuscript has been deemed suitable for publication in PLOS ONE. Congratulations! Your manuscript is now with our production department. 

Kind regards, 

on behalf of

Dr. Amir Radfar 

Academic Editor

PLOS ONE